# Equivariant Spatio-Temporal Attentive Graph Networks to Simulate Physical Dynamics

**Liming Wu**[1,2]\*, **Zhichao Hou**[3]\*, **Jirui Yuan**[4], **Yu Rong**[5], **Wenbing Huang**[1,2]†

[1]Gaoling School of Artificial Intelligence, Renmin University of China
[2]Beijing Key Laboratory of Big Data Management and Analysis Methods, Beijing, China
[3]Department of Computer Science, North Carolina State University
[4]Institute for AI Industry Research (AIR), Tsinghua University   [5]Tencent AI Lab
{manliowu,hwenbing}@ruc.edu.cn zhou4@ncsu.edu
yu.rong@hotmail.com yuanjirui@air.tsinghua.edu.cn

## Abstract

Learning to represent and simulate the dynamics of physical systems is a crucial yet challenging task. Existing equivariant Graph Neural Network (GNN) based methods have encapsulated the symmetry of physics, *e.g.*, translations, rotations, etc, leading to better generalization ability. Nevertheless, their frame-to-frame formulation of the task overlooks the non-Markov property mainly incurred by unobserved dynamics in the environment. In this paper, we reformulate dynamics simulation as a spatio-temporal prediction task, by employing the trajectory in the past period to recover the Non-Markovian interactions. We propose Equivariant Spatio-Temporal Attentive Graph Networks (ESTAG), an equivariant version of spatio-temporal GNNs, to fulfill our purpose. At its core, we design a novel Equivariant Discrete Fourier Transform (EDFT) to extract periodic patterns from the history frames, and then construct an Equivariant Spatial Module (ESM) to accomplish spatial message passing, and an Equivariant Temporal Module (ETM) with the forward attention and equivariant pooling mechanisms to aggregate temporal message. We evaluate our model on three real datasets corresponding to the molecular-, protein- and macro-level. Experimental results verify the effectiveness of ESTAG compared to typical spatio-temporal GNNs and equivariant GNNs.

## 1 Introduction

It has been a goal to represent and simulate the dynamics of physical systems by making use of machine learning techniques [37, 8, 12]. The related studies, once pushed forward, have great potential to facilitate a variety of downstream scientific tasks including Molecular Dynamic (MD) simulation [19], protein structure prediction [1], virtual screening of drugs and materials [30], model-based robot planning/control [35], and many others.

Plenty of solutions have been proposed, amongst which the usage of Graph Neural Networks (GNNs) [41] becomes one of the most desirable directions. GNNs naturally model particles or unit elements as nodes, physical relations as edges, and the latent interactions as the message passing thereon. More recently, a line of researches [38, 11, 21, 9, 32, 20] have been concerned with generalizing GNNs to fit the symmetry of our physical world. These works, also known as equivariant GNNs [18], ensure that translating/rotating/reflecting the geometric input of GNNs results in the output transformed in the same way. By handcrafting such Euclidean equivariance, the

---

\*Equal contribution.
†Corresponding author.

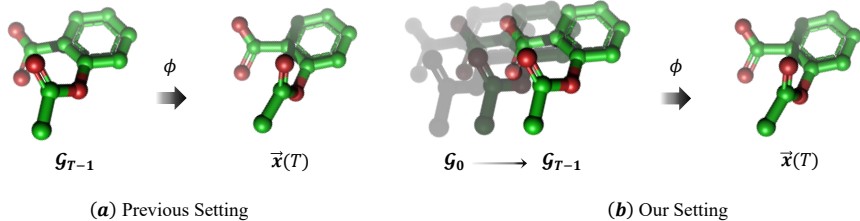

(**a**) Previous Setting                    (**b**) Our Setting

Figure 1: Comparison of the problem setting between previous methods and our paper. Here, we choose the dynamics of the Aspirin molecular with time lag as 1 for illustration.

models are well predictable to scenarios under arbitrary coordinate systems, giving rise to enhanced generalization ability.

In spite of the fruitful progress, existing methods overlook a vital point: *the observed physical dynamics are almost non-Markovian*. In previous methods, they usually take as input the conformation of a system at a single temporal frame and predict as output the future conformation after a fixed time interval, forming a frame-to-frame forecasting problem. Under the Markovian assumption, this setting is pardonable as future frames are independent of all other past frames given the input one. Nevertheless, the Markovian assumption is rather unrealistic when there are other unobserved objects interacting with the system we are simulating [40]. For instance, when we consider simulating the dynamics of a protein that is interacting with an unobserved solvent (such as water), the Markovian property no longer holds; in other words, even conditional on the current frame, the future dynamics of the protein depends on the current state of the solvent which, however, is influenced by the past states of the protein itself, owing to the interaction between the protein and the solvent. The improper Markovian assumption makes current works immature in dynamics modeling.

To relieve from the Markovian assumption, this paper proposes to employ the states in the past period to reflect the latent and unobserved dynamics. In principle, we can recover the non-Markovian behavior (*e.g.* interacting with a solvent) if the past period is sufficiently long. We collect a period of past system states as spatio-temporal graphs, and utilize them as the input to formulate a spatio-temporal prediction task, other than the frame-to-frame problem as usual (see Figure 1). This motivates us to leverage existing Spatio-Temporal GNNs (STGNNs) [44] to fulfill our purpose, which, unfortunately, are unable to conform to the aforementioned Euclidean symmetry and the underlying physical laws. Hence, the equivariant version of STGNNs is no doubt in demand. Another point is that periodic motions are frequently observed in typical physical systems [3]. For example, the Aspirin molecular exhibits clear periodic thermal vibration when binding to a target protein. Under the spatio-temporal setting, we are able to model periodicity of dynamics, which, however, is less investigated before.

Our contributions are summarized as follows:

- We reveal the non-Markov behavior in physical dynamics simulation by developing equivariant spatio-temporal graph models. The proposed model dubbed Equivariant Spatio-Temporal Attentive Graph network (ESTAG) conforms to Euclidean symmetry and alleviates the limitation of the Markovian assumption.

- We design a novel Equivariant Discrete Fourier Transform (EDFT) to extract periodic features from the dynamics, and then construct an Equivariant Spatial Module (ESM), and an Equivariant Temporal Module (ETM) with forward attention and equivariant pooling, to process spatial and temporal message passing, respectively.

- The effectiveness of ESTAG is verified on three real datasets corresponding to the molecular-, protein- and macro-level. We prove that, involving both temporal memory and equivariance are advantageous compared to typical STGNNs and equivariant GNNs with adding trivial spatio-temporal aggregation.

## 2   Related Work

**GNNs for Physical Dynamics Modeling** Graph Neural Networks (GNNs) have shown great potential in physical dynamics modeling. IN [2], NRI [23], and HRN [29] are a series of works to learn physical

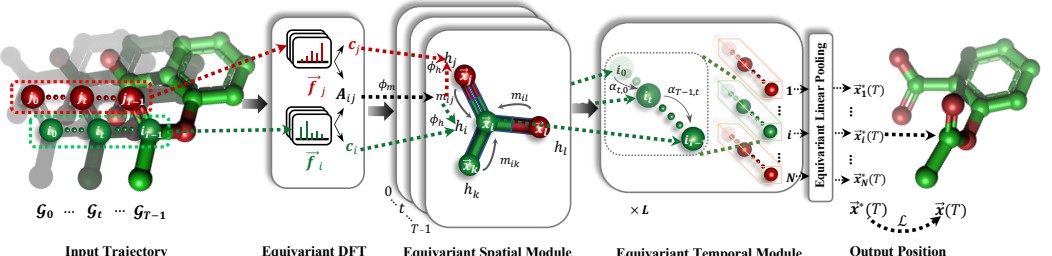

Figure 2: Schematic overview of ESTAG. After inputting historical graph trajectories $\mathcal{G}_0, ..., \mathcal{G}_{T-1}$, Equivariant Discrete Fourier Transform (EDFT) extracts equivariant frequency features $\vec{f}$ from the trajectory. We process them into the invariant node-wise feature $c$ and adjacency matrix $A$ to be adopted for the next stage. Then we stack Equivariant Spatial Module (ESM) and Equivariant Temporal Module (ETM) alternatively for $L$ times to explore spatial and temporal dependencies. After the equivariant temporal pooling layer, we obtain the estimated position $\vec{x}^*(T)$.

system interaction and evolution. Considering the energy conservation and incorporating physical prior knowledge into GNNs, HNN [15], HOGN [31] leverage ODE and Hamiltonian mechanics to capture the interactions in the systems. However, all of the above-mentioned models don't take into account the underlying symmetries of a system. In order to introduce Euclidean equivariance, Tensor-Field networks (TFN) [38] and SE(3)-Transformer [11] equip filters with rotation equivariance by irreducible representation of the SO(3) group. LieTransformer [21]and LieConv [9] leverage the Lie group to enforce equivariance. Besides, EGNN [32] utilized a simpler E(n)-equivariant framework which can achieve competitive results without computationally expensive information. Based on EGNN, GMN [20] further proposes an equivariant and constraint-aware architecture by making use of forward kinematics information in physical system. Nevertheless, all of these methods ignore the natural spatiotemporal patterns of physical dynamics and model it as a frame-to-frame forecasting problem.

**Spatio-temporal graph neural networks** STGNNs [22] aim to capture the spatial and temporal dependency simultaneously and are widely investigated in various applications like traffic forecasting and human recognition. DCRNN [26] and GaAN [45] are RNN-based methods which filter inputs and hidden states passed to recurrent unit using graph convolutions. However, RNN-based approaches are time-consuming and may suffer from gradient explosion/vanishing problem. CNN-based approaches, such as STGCN [44] and ST-GCN [43], interleave 1D-CNN layers with graph convolutional layers to tackle spatial temporal graphs in a non-recursive manner. Besides, attention mechanism, an important technique STGCNs, is employed by GaAN [45], AGL-STAN [36] and ASTGCN [16] to learn dynamic dependencies in both space and time domain. The aforementioned approaches are targeted to the applications in 2D graph scenarios such as traffic networks and human skeleton graphs, and may not be well applicable to 3D physical systems in which geometric equivariance is a really important property.

**Equivariant spatio-temporal graph neural networks** There are few previous works designing spatio-temporal GNNs while maintaining equivariance. Particularly, by using GRU [6] to record the memory of past frames, LoCS [24] additionally incorporates rotation-invariance to improve the model's generalization ability. Different from the recurrent update mechanism used in LoCs, EqMotion [42] distills the history trajectories of each node into a multi-dimension vector, by which the spatio-temporal graph is compressed as a spatial graph, then it designs an equivariant module and an interaction reasoning module to predict future frames. However, both of LoCS and EqMotion are still defective in exploring the interactions among history trajectories, while in this paper we propose a transformer-alike architecture to fully leverage the spatio-temporal interactions based on equivariant attentions.

## 3 Notations and Task Definition

The dynamics of physical objects (such as molecules) can be formulated with the notion of spatiotemporal graphs, as shown in Figure 1 (left). In particular, a spatiotemporal graph of node number $N$ and

temporal length $T$ is denoted as $\{\mathcal{G}_t = (\mathcal{V}_t, \mathcal{E})\}_{t=0}^{T-1}$. Here, the nodes $\mathcal{V}_t$ are shared across time, but each node $i$ is assigned with different scalar feature $\boldsymbol{h}_i(t) \in \mathbb{R}^c$, position vector $\vec{\boldsymbol{x}}_i(t) \in \mathbb{R}^3$ at different temporal frame $t$; the edges $\mathcal{E}$ are associated with an identical adjacency matrix $\boldsymbol{A} \in \mathbb{R}^{N \times N \times m}$ whose element $\boldsymbol{A}_{ij} \in \mathbb{R}^m$ defines the edge feature between node $i$ and $j$. We henceforth denote by the matrices $\boldsymbol{H}(t) \in \mathbb{R}^{c \times N}$ and $\vec{\boldsymbol{X}}(t) \in \mathbb{R}^{3 \times N}$ the collection of all nodes in $\mathcal{G}_t$. Here for simplicity, we specify the time lag as $\Delta t = 1$ which could be valued more than 1 in practice. In general, the $t$-th frame corresponds to time $T - t\Delta t$.

**Task Definition** This paper is interested in predicting the physical state, particularly the position of each node at frame $T$ given the historical graph series $\{\mathcal{G}_t\}_{t=0}^{T-1}$. In form, we learn the function $\phi$:

$$\{(\boldsymbol{H}(t), \vec{\boldsymbol{X}}(t), \boldsymbol{A})\}_{t=0}^{T-1} \xrightarrow{\phi} \vec{\boldsymbol{X}}(T). \tag{1}$$

Although previous approaches such as EGNN [32] and GMN [20] also claim to tackle physical dynamics modeling, they neglect the application of spatiotemporal patterns and only accomplish frame-to-frame prediction, where the input of $\phi$ is reduced to a single frame, *e.g.*, $\mathcal{G}_{T-1}$ in Eq. 1.

**Equivariance** A crucial constraint on physical dynamics is that the function $\phi$ should meet the symmetry of our 3D world. In other words, for any translation/rotation/reflection $g$ in the group E(3), $\phi$ satisfies:

$$\phi(\{(\boldsymbol{H}(t), g \cdot \vec{\boldsymbol{X}}(t), \boldsymbol{A})\}_{t=0}^{T-1}) = g \cdot \vec{\boldsymbol{X}}(T), \tag{2}$$

where the group action $\cdot$ is instantiated as $g \cdot \vec{\boldsymbol{X}}(t) := \vec{\boldsymbol{X}}(t) + \boldsymbol{b}$ for translation $\boldsymbol{b} \in \mathbb{R}^3$ and $g \cdot \vec{\boldsymbol{X}}(t) := \boldsymbol{O}\vec{\boldsymbol{X}}(t)$ for rotation/reflection $\boldsymbol{O} \in \mathbb{R}^{3 \times 3}$.

# 4  Our approach: ESTAG

Discovering informative spatiotemporal patterns from the input graph series is vital to physical dynamics modeling. In this section, we introduce ESTAG that pursues this goal in a considerate way: We first extract the frequency of each node's trajectory by EDFT (§ 4.1), which captures the node-wise temporal dynamics in a global sense and returns important features for the next stage; We then separately characterize the spatial dependency among the nodes of each input graph $\mathcal{G}_t$ via ESM (§ 4.2); We finally unveil the temporal dynamics of each node through the attention-based mechanism ETM and output the estimated position of each node in $\mathcal{G}_T$ after an equivariant temporal pooling layer (§ 4.3). The overall architecture is shown in Figure 2.

## 4.1  Equivariant Discrete Fourier Transform (EDFT)

The Fourier Transform (FT) gives us insight into the wave frequencies contained in the input signal that is usually periodic. With the extracted frequencies, we are able to view the global behavior of each node $i$ in different frequency domains. Conventional multidimensional FT employs distinct Fourier bases for different input dimensions of the original signals. Here, to ensure equivariance, we first translate the signals by the mean position and then adopt the same basis over the spatial dimension. To be specific, we compute equivariant DFT as follows:

$$\vec{\boldsymbol{f}}_i(k) = \sum_{t=0}^{T-1} e^{-i' \frac{2\pi}{T} kt} \left( \vec{\boldsymbol{x}}_i(t) - \overline{\vec{\boldsymbol{x}}(t)} \right), \tag{3}$$

where, $i'$ is the imaginary unit, $k = 0, 1, \cdots, T-1$ is the frequency index, $\overline{\vec{\boldsymbol{x}}(t)}$ is the average position of all nodes in the $t$-th frame $\mathcal{G}_t$, and the output $\vec{\boldsymbol{f}}_i(k) \in \mathbb{C}^3$ is complex. The frequencies calculated by Eq. 3 are then utilized to formulate two crucial quantities: the frequency cross-correlation $\boldsymbol{A}_{ij} \in \mathbb{R}^T$ between node $i$ and $j$, and the frequency amplitude $\boldsymbol{c}_i \in \mathbb{R}^T$ of node $i$.

In signal processing, cross-correlation measures the similarity of two functions $f_1$ and $f_2$. It satisfies $\mathcal{F}\{f_1 \star f_2\} = \overline{\mathcal{F}\{f_1\}} \cdot \mathcal{F}\{f_2\}$, where $\mathcal{F}$ and $\star$ denote the FT and the cross-correlation operator, respectively, and $\overline{\mathcal{F}}$ indicates the complex conjugate of $\mathcal{F}$. Borrowing this idea, we compute the cross-correlation in the frequency domain by Eq. 3 as:

$$\boldsymbol{A}_{ij}(k) = w_k(\boldsymbol{h}_i)w_k(\boldsymbol{h}_j)|\langle \vec{\boldsymbol{f}}_i(k), \vec{\boldsymbol{f}}_j(k)\rangle|, \tag{4}$$

where $\langle \cdot, \cdot \rangle$ defines the complex inner product. Notably, we have added two learnable parameters $w_k(\boldsymbol{h}_i)$ and $w_k(\boldsymbol{h}_j)$ dependent on node features, which act like spectral filters of the $k$-th frequency and enable us to select related frequency for the prediction. In the next subsection, we will apply $\boldsymbol{A}_{ij}$ as the edge feature to capture the relationship between different nodes. We use Aspirin as an example and visualize the $\boldsymbol{A}$ in Figure 3.

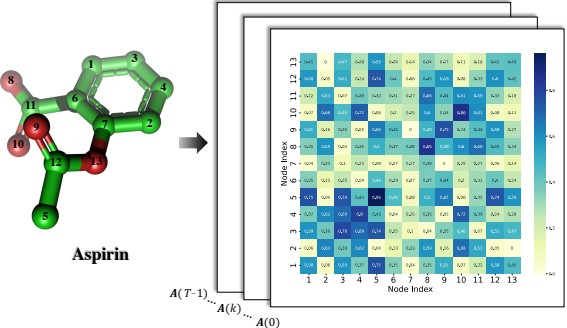

Figure 3: Visualization of cross-correlation $\boldsymbol{A}$ on Aspirin. EDFT can not only identify strongly-connected nodes (e.g. Node 8 and Node 11), but also discover latent relationship between two nodes which are disconnected yet may have similar structures or functions (e.g. Node 8 and Node 10).

We further compute for node $i$ the amplitude of the frequency $\vec{\boldsymbol{f}}_i(k)$ along with the parameter $w_k(\boldsymbol{h}_i)$:

$$\boldsymbol{c}_i(k) = w_k(\boldsymbol{h}_i)\|\vec{\boldsymbol{f}}_i(k)\|^2. \tag{5}$$

This term will be used in the update of the hidden features in the next subsection.

A promising property of Eq. 3 is that it is translation invariant and rotation/reflection equivariant. Therefore, both $\boldsymbol{A}_{ij}$ and $\boldsymbol{c}_i$ are E(3)-invariant, which will facilitate the design of following modules.

### 4.2 Equivariant Spatial Module (ESM)

For each graph $\mathcal{G}_t$, our ESM is proposed to encode its spatial geometry through equivariant message passing. ESM is built upon EGNN [33] which is a prevailing kind of equivariant GNNs, but it has subtly involved the FT features from the last subsection for enhanced performance beyond EGNN.

The $l$-th layer message passing in ESM is as below:

$$\boldsymbol{m}_{ij} = \phi_m \left( \boldsymbol{h}_i^{(l)}(t), \boldsymbol{h}_j^{(l)}(t), \|\vec{\boldsymbol{x}}_{ij}^{(l)}(t)\|^2, \boldsymbol{A}_{ij} \right), \tag{6}$$

$$\boldsymbol{h}_i^{(l+1)}(t) = \boldsymbol{h}_i^{(l)}(t) + \phi_h \left( \boldsymbol{h}_i^{(l)}(t), \boldsymbol{c}_i, \sum_{j \neq i} \boldsymbol{m}_{ij} \right), \tag{7}$$

$$\vec{\boldsymbol{a}}_i(t) = \frac{1}{|\mathcal{N}(i)|} \sum_{j \in \mathcal{N}(i)} \vec{\boldsymbol{x}}_{ij}^{(l)}(t)\phi_x(\boldsymbol{m}_{ij}), \tag{8}$$

$$\vec{\boldsymbol{x}}_i^{(l+1)}(t) = \vec{\boldsymbol{x}}_i^{(l)}(t) + \vec{\boldsymbol{a}}_i(t), \tag{9}$$

where, $\phi_m$ computes the message $\boldsymbol{m}_{ij}$ from node $j$ to $i$, $\phi_h$ updates the hidden representation $\boldsymbol{h}_i$, $\phi_x$ returns a one-dimensional scalar for the update of $\vec{\boldsymbol{a}}_i(t)$, and all the above functions are Multi-Layer Perceptrons (MLPs); $\vec{\boldsymbol{x}}_{ij}(t) = \vec{\boldsymbol{x}}_i(t) - \vec{\boldsymbol{x}}_j(t)$ is the relative position and $\mathcal{N}(i)$ denotes the neighborhoods of node $i$.

Notably, we leverage the cross-correlation $\boldsymbol{A}_{ij}$ as the edge feature in Eq. 6 to evaluate the connection between node $i$ and $j$ over the global temporal window, since it is computed from the entire trajectory. We also make use of $\boldsymbol{c}_i$ as the input of the update in Eq. 7. The benefit of considering these two terms will be ablated in our experiments.

### 4.3 Equivariant Temporal Module (ETM)

**Forward Temporal Attention** Inspired by the great success of Transformer [39] in sequence modeling, we develop ETM that describes the self-correspondence of each node's trajectory based on the foward attention mechanism, and more importantly, in an E(3)-equivariant way.

In detail, each layer of ETM conducts the following process:

$$\alpha_i^{(l)}(ts) = \frac{\exp(\boldsymbol{q}_i^{(l)}(t)^\top \boldsymbol{k}_i^{(l)}(s))}{\sum_{s=0}^t \exp(\boldsymbol{q}_i^{(l)}(t)^\top \boldsymbol{k}_i^{(l)}(s))}, \tag{10}$$

$$\boldsymbol{h}_i^{(l+1)}(t) = \boldsymbol{h}_i^{(l)}(t) + \sum_{s=0}^t \alpha_i^{(l)}(ts)\boldsymbol{v}_i^{(l)}(s), \tag{11}$$

$$\vec{\boldsymbol{x}}_i^{(l+1)}(t) = \vec{\boldsymbol{x}}_i^{(l)}(t) + \sum_{s=0}^t \alpha_i^{(l)}(ts)\vec{\boldsymbol{x}}_i^{(l)}(ts)\phi_x(\boldsymbol{v}_i^{(l)}(s)), \tag{12}$$

where, $\alpha_{ts}$ is the attention weight between time $t$ and $s$, computed by the query $\boldsymbol{q}_t$ and key $\boldsymbol{k}_s$; the hidden feature $\boldsymbol{h}_i(t)$ is updated as a weighted combination of the value $\boldsymbol{v}_s$; the position vector $\vec{\boldsymbol{x}}_i(t)$ is derived from a weighted combination of a one-dimensional scalar $\phi_x(\boldsymbol{v}_s)$ multiplied with the temporal displace vector $\vec{\boldsymbol{x}}_i^{(l)}(ts) = \vec{\boldsymbol{x}}_i^{(l)}(t) - \vec{\boldsymbol{x}}_i^{(l)}(s)$. Specifically, $\boldsymbol{q}_i^{(l)}(t) = \phi_q\left(\boldsymbol{h}_i^{(l)}(t)\right)$, $\boldsymbol{k}_i^{(l)}(t) = \phi_k\left(\boldsymbol{h}_i^{(l)}(t)\right)$ and $\boldsymbol{v}_i^{(l)}(t) = \phi_v\left(\boldsymbol{h}_i^{(l)}(t)\right)$ are all E(3)-invariant functions. Notably, we derive a particle's next position in a forward-looking way, to keep physical rationality as the derivation of current state should not be dependent on future positions.

**Equivariant Temporal Pooling** We alternate one-layer ESM and one-layer ETM over $L$ layers, and finally attain the updated coordinates $\vec{\boldsymbol{x}}_i^{(L)} \in \mathbb{R}^{T \times 3}$ for each node $i$. Then the predicted coordinates at time $T$ is given by the following equivariant linear pooling:

$$\vec{\boldsymbol{x}}_i^*(T) = \hat{\boldsymbol{X}}_i \boldsymbol{w} + \vec{\boldsymbol{x}}_i^{(L)}(T-1), \tag{13}$$

where the parameter $\boldsymbol{w} \in \mathbb{R}^{(T-1)}$ consists of learnable weights, and $\hat{\boldsymbol{X}}_i = [\vec{\boldsymbol{x}}_i^{(L)}(0) - \vec{\boldsymbol{x}}_i^{(L)}(T-1), \vec{\boldsymbol{x}}_i^{(L)}(1) - \vec{\boldsymbol{x}}_i^{(L)}(T-1), \cdots, \vec{\boldsymbol{x}}_i^{(L)}(T-2) - \vec{\boldsymbol{x}}_i^{(L)}(T-1)]$ is translated by $\vec{\boldsymbol{x}}_i^{(L)}(T-1)$ to allow translation invariance.

We train ESTAG end-to-end via the mean squared error (MSE) loss:

$$\mathcal{L} = \sum_{i=1}^N \|\vec{\boldsymbol{x}}_i(T) - \vec{\boldsymbol{x}}_i^*(T)\|_2^2. \tag{14}$$

By the design of EDFT, ESM and ETM, we have the following property of our model ESTAG.

**Theorem 4.1.** *We denote ESTAG as* $\vec{\boldsymbol{X}}(T) = \phi\left(\{(\boldsymbol{H}(t), g \cdot \vec{\boldsymbol{X}}(t), \boldsymbol{A})\}_{t=0}^{T-1}\right)$, *then* $\phi$ *is E(3)-equivariant.*

*Proof. See Appendix A.*

Although we mainly exploit EGNN as the backbone (particularly in ESM), our framework is general and can be easily extended to other equivariant GNNs, such as GMN [20], the multi-channel version of EGNN. In general, the extended models deal with multi-channel coordinate $\vec{\boldsymbol{Z}} \in \mathbb{R}^{3 \times m}$ instead of $\vec{\boldsymbol{x}} \in \mathbb{R}^3$. The most significant feature of these models is to replace the invariant scalar $\|\vec{\boldsymbol{x}}\|^2$ in the formulations of the message $\boldsymbol{m}_{ij}$ (Eq. 6) with the term $\vec{\boldsymbol{Z}}^\top \vec{\boldsymbol{Z}}$. It is easily to prove that this term is equivariant to any orthogonal matrix $\boldsymbol{O}$, *i.e.*, $(\boldsymbol{O}\vec{\boldsymbol{Z}})^\top(\boldsymbol{O}\vec{\boldsymbol{Z}}) = \vec{\boldsymbol{Z}}^\top \vec{\boldsymbol{Z}}, \forall \boldsymbol{O} \in \mathbb{R}^{3 \times 3}, \boldsymbol{O}^\top \boldsymbol{O} = \boldsymbol{I}$. Besides, it can be reduced to invariant scalar $\|\vec{\boldsymbol{x}}\|^2$ when $m = 1$. Empirically, we add the normalization term in order to achieve more stable performance: $\frac{\vec{\boldsymbol{Z}}^\top \vec{\boldsymbol{Z}}}{\|\vec{\boldsymbol{Z}}^\top \vec{\boldsymbol{Z}}\|_F}$, where $\|\cdot\|_F$ is the Frobenius norm. In the above derivation, we only display the formulation on EGNN, the details of multi-channel ESTAG are shown in Appendix C.1.

Table 1: Prediction error ($\times 10^{-3}$) on MD17 dataset. Results averaged across 3 runs. We do not display the standard deviation due to its small value.

| | ASPIRIN | BENZENE | ETHANOL | MALONALDEHYDE | NAPHTHALENE | SALICYLIC | TOLUENE | URACIL |
|---|---|---|---|---|---|---|---|---|
| PT-$s$ | 15.579 | 4.457 | 4.332 | 13.206 | 8.958 | 12.256 | 6.818 | 10.269 |
| PT-$m$ | 9.058 | 2.536 | 2.688 | 6.749 | 6.918 | 8.122 | 5.622 | 7.257 |
| PT-$t$ | 0.715 | 0.114 | 0.456 | 0.596 | 0.737 | 0.688 | 0.688 | 0.674 |
| EGNN-$s$ | 12.056 | 3.290 | 2.354 | 10.635 | 4.871 | 8.733 | 3.154 | 6.815 |
| EGNN-$m$ | 6.237 | 1.882 | 1.532 | 4.842 | 3.791 | 4.623 | 2.516 | 3.606 |
| EGNN-$t$ | 0.625 | 0.112 | 0.416 | 0.513 | 0.614 | 0.598 | 0.577 | 0.568 |
| ST_TFN | 0.719 | 0.122 | 0.432 | 0.569 | 0.688 | 0.684 | 0.628 | 0.669 |
| ST_GNN | 1.014 | 0.210 | 0.487 | 0.664 | 0.769 | 0.789 | 0.713 | 0.680 |
| ST_SE(3)TR | 0.669 | 0.119 | 0.428 | 0.550 | 0.625 | 0.630 | 0.591 | 0.597 |
| ST_EGNN | 0.735 | 0.163 | 0.245 | 0.427 | 0.745 | 0.687 | 0.553 | 0.445 |
| EQMOTION | 0.721 | 0.156 | 0.476 | 0.600 | 0.747 | 0.697 | 0.691 | 0.681 |
| STGCN | 0.715 | 0.106 | 0.456 | 0.596 | 0.736 | 0.682 | 0.687 | 0.673 |
| AGL-STAN | 0.719 | 0.106 | 0.459 | 0.596 | 0.601 | 0.452 | 0.683 | 0.515 |
| ESTAG | **0.063** | **0.003** | **0.099** | **0.101** | **0.068** | **0.047** | **0.079** | **0.066** |

## 5 Experiments

**Datasets.** To verify the superiority of the proposed model, we evaluate our model on three real world datasets: **1)** molecular-level: MD17 [5], **2)** protein-level: AdK equilibrium trajectory dataset [34] and **3)** macro-level: CMU Motion Capture Databse [7]. These datasets involve several continuous long trajectories. Note that all the three datasets contain unobserved dynamics or factors and thus conform to the non-Markovian setting. In particular, the external temperature and pressure are unknown on MD17, the dynamics of water and ions is unobserved on AdK, and the states of the environment are not provided on Motion Capture. The original datasets are composed of long trajectories. We randomize the start point and extract the following $T + 1$ points with the interval $\Delta t$. We take the first $T$ timestamps as previous observations and last timestamp as the future position label.

**Baselines.** We compare the performance of ESTAG with several baselines: **1)** We regard the previous observation at start/mid/terminal ($s/m/t$) timepoint as the estimated position at timestamp $T$ directly. **2) EGNN** [33] utilizes a simple yet efficient framework which transforms the 3D vectors into invariant scalars. We provide EGNN with only one previous position at $s/m/t$ timepoint to predict the future position in a frame-to-frame manner. **3) STGCN** [44] is a spatio-temporal GNN that adopts a "sandwich" structure with two gated sequential convolution layers and on spatial graph convolution layer in between. We modify its default settings by predicting the residual coordinate between time $T - 1$ and $T$, since directly predicting the exact coordinate at time $T$ yields much worse performance.**4) AGL-STAN** [36] leverages adaptive graph learning and self-attention for a comprehensive representation of intra-temporal dependencies and inter-spatial interactions. We modify AGL-STAN's setting in the same way as we do with STGCN. **5)** Typical GNNs with trivial spatio-temporal aggregation: we implement GNN [13], and other equivariant models EGNN, TFN [38], and SE(3)-Transformer [11] for each temporal frame in the historical trajectory and then estimate the future position as the weighted sum of all past frames, where the weights are learnable. All models are denoted with a prefix "ST" and we initialize their node features along with temporal positional encoding. **6) EqMotion** [42] is one of equivariant spatio-temporal GNNs, which leverages the temporal information by fusing them for the model's initialization.

### 5.1 Molecular-level: MD17

**Implementation details.** MD17 dataset includes the trajectories of 8 small molecules generated by MD simulation. We use the atomic number as the time-independent input node feature $h^{(0)}$. The two atoms are 1-hop neighbor if their distance is less than the threshold $\lambda$ and we consider two types of neighbors (i.e. 1-hop neighbor and 2-hop neighbor). Other settings including the hyper-parameters are introduced in Appendix D.1.

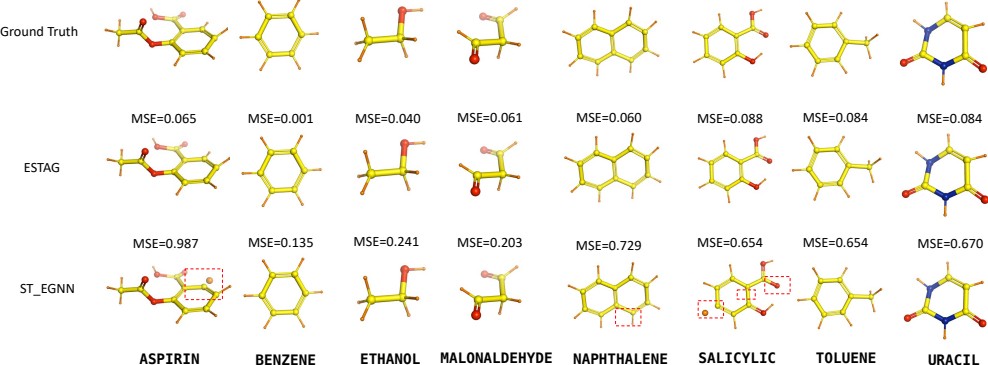

Figure 4: PyMol visualization of the predicted molecules by our ESTAG and ST_EGNN, where the MSE ($\times 10^{-3}$) with respect to the ground truth is also shown. As expected, the predicted instances by ESTAG exhibit much smaller MSE than ST_EGNN, although the difference is not easy to visualize in some cases. For those obviously mispredicted regions of ST_EGNN, we highlight them with red rectangles. It is observed that ST_EGNN occasionally outputs isolated atoms, which could be caused by violation of the bond length tolerance in PyMol.

**Results.** Table 1 shows the average MSE of all models on 8 molecules. We have some interesting observations as follows: **1)** ESTAG exceeds other models in all cases by a significant margin, supporting the general effectiveness of the proposed ideas. **2)** From the Pt-$s/m/t$, we observe that the point closer to the future point has less prediction error. EGNN-$s/m/t$ only takes one frame ($s/m/t$) as input, and attains slight improvement relative to Pt-$s/m/t$. **3)** Compared with EGNN-$t$, ST_EGNN, although equipped with trivial spatio-temporal aggregation, is unable to obtain consistent improvement, which indicates that how to unveil temporal dynamics appropriately is beyond triviality on this dataset. **4.** The non-equivariant methods particularly ST_GNN perform unsatisfactorily in most cases, implying that equivariance is an important property when modeling 3D structures.

**Visualization.** We have provided some visualizations of the predicted molecules by using the PyMol toolkit. Figure 4 shows that the prdicted MSEs of our model are much lower than ST_EGNN and our predicted molecules are closer to the ground-truth molecules. We have also displayed the learned attentions (Eq. 10) and the temporal weight (Eq. 13), where we find meaningful patterns of the temporal correlation. For clearer comparison, we display the molecule URACIL in 3D coordinate system, which can be found in Appendix D.5.

## 5.2 Protein-level: Protein Dynamics

**Implementation details.** We evaluate our model on the AdK equilibrium trajectory dataset [34] via MDAnalysis toolkit [14]. In order to reduce the data scale, we utilize MDAnalysis to locate the backbone atoms ($C_\alpha, C, N, O$) of the residues and then regard the residues other than atoms in protein as the nodes with $4$-channel geometric features. We use the atomic number of four backbone atoms as the time-independent input node feature $h^{(0)}$. We connect two atoms via an edge if their distance is less than a threshold $\lambda$. Other settings including the hyperparameters are introduced in Appendix D.2. Slightly different from the model on MD17 dataset, we generalize the single-channel ESTAG into multi-channel version which is presented in detail in Appendix C. We do not conduct EqMotion, TFN and SE(3)-TR used in the last experiment since it is non-trivial to modify them into multi-channel modeling. The state-of-the-art method GMN [20] is implemented by further adding the weighed temporal pooling similar to ST_EGNN.

Table 2: Prediction error and training time on Protein dataset. Results averaged across 3 runs.

| METHOD | MSE | TIME(S) |
|---|---|---|
| PT-$s$ | 3.260 | - |
| PT-$m$ | 3.302 | - |
| PT-$t$ | 2.022 | - |
| EGNN-$s$ | 3.254 | 1.062 |
| EGNN-$m$ | 3.278 | 1.088 |
| EGNN-$t$ | 1.983 | 1.069 |
| ST_GNN | 1.871 | 2.769 |
| ST_GMN | 1.526 | 4.705 |
| ST_EGNN | 1.543 | 4.705 |
| STGCN | 1.578 | 1.840 |
| AGL-STAN | 1.671 | 1.478 |
| ESTAG | **1.471** | 6.876 |

**Results.** The predicted MSEs are displayed in Table 2. Generally, the spatio-temporal models are better than Pt-$s/m/t$ and EGNN-$s/m/t$, and it suggests that applying spatio-temporal clues on this dataset is crucial, particularly given that the protein trajectories are generated under the interactions with external molecules such as water and ions. The equivariant models always outperform the non-equivariant counterparts (for example, ST_EGNN vs. ST_GNN). Overall, our model ESTAG achieves the best performance owing to its elaboration of equivariant spatio-temporal modeling. Additionally, we report the training time averaged over epochs in Table 2. It shows that the computation overhead of ESTAG over its backbone EGNN is acceptable given its remarkable performance enhancement. It is expected that the superiority of ESTAG on protein dataset is not as obvious as that on MD17, owing to various kinds of physical interactions between different amino acids, let along each amino acid is composed of a certain number of atoms, which makes the dynamics of a protein much more complicated than small molecules.

### 5.3 Macro-level: Motion Capture

**Implementation details.** We finally adopt CMU Motion Capture Database [7] to evaluate our model. CMU Motion Capture Database involves the trajectories of human motion under several scenarios and we focus on walking motion (subject #35) and basketball motion (subject #102, only take trajectories whose length is greater than 170). The input feature of all the joints (nodes) $h_i^{(0)}$ are all 1s. The two joints are 1-hop neighbor if they are connected naturally and we consider two types of neighbors (i.e. 1-hop neighbor and 2-hop neighbor). Other settings including the hyper-parameters are introduced in Appendix D.3. Notably, the input of EqMotion only contain node coordinates, as the same as our method and other baselines. We find that EqMotion performs much worse by directly predicting the absolute coordinates. We then modify EqMotion to predict the relative coordinates across two adjacent frames and perform zero-mean normalization of node coordinates, for further improvement.

Table 3: Prediction error ($\times 10^{-1}$) on Motion dataset. Results averaged across 3 runs.

| METHOD | WALK | BASKETBALL |
|---|---|---|
| PT-$s$ | 329.474 | 886.023 |
| PT-$m$ | 127.152 | 413.306 |
| PT-$t$ | 3.831 | 15.878 |
| EGNN-$s$ | 63.540 | 749.486 |
| EGNN-$m$ | 32.016 | 335.002 |
| EGNN-$t$ | 0.786 | 12.492 |
| ST_GNN | 0.441 | 15.336 |
| ST_TFN | 0.597 | 13.709 |
| ST_SE(3)TR | 0.236 | 13.851 |
| ST_EGNN | 0.538 | 13.199 |
| EQMOTION | 1.011 | 4.893 |
| STGCN | 0.062 | 4.919 |
| AGL-STAN | **0.037** | 5.734 |
| ESTAG | 0.040 | **0.746** |

**Results.** Table 3 summarizes the results of all models on the Motion dataset. The spatio-temporal models are better than Pt-$s/m/t$ and EGNN-$s/m/t$, which again implies the necessity of taking the spatio-temporal history into account. Unexpectedly, the non-equivariant models are even superior to the equivariant baselines in walking motion, by, for instance, comparing AGL-STAN with ESTAG. We conjure that the samples of this dataset are usually collected in the same orientation, which potentially subdues the effect of rotation equivariance. It is thus not surprising that GNN even outperforms EGNN since GNN involves more flexible form of message passing. But for basketball motion which is more complicated to simulate, ESTAG yields a much lower MSE. We also notice that the attention-based models including our ESTAG, ST_SE(3)-Tr, and AGL-STAN perform promisingly, which probably due to the advantage of using attention to discovery temporal interactions within the trajectories.

### 5.4 Ablation Studies

Here we conduct several ablation experiments on MD17 to inspect how our proposed components contribute to the overall performance and the results are shown in Table 4.

**1)** Without EDFT. We replace the FT-based edge features with the predefined edge features based on the connecting atom types and the distance between them. The simplified model encounters an average of increase in MSE, showcasing the effectiveness of the FT-based feature in modeling the spatial relation in graphs.

**2)** Without attention. We remove the ETM of ESTAG and observe slight detriment in the model performance, which demonstrates that attention mechanism can well capture the temporal dynamics.

**3)** Without equivariance. We construct a non-equivariant spatio-temporal attentive framework based on vanilla GNN. ESTAG performs much better than the non-equivariant STAG, indicating that Euclidean equivariance is a crucial property when designing model on geometric graph.

**4)** The impact of the number of layers $L$. We investigate effect of the number of layers $L$ on Ethanol dataset. We vary $T$ from 1, 2, 3, 4, 5, 6 and present the results in Table 5. Considering the accuracy and efficiency simultaneously, we choose $L = 2$ for the ESTAG.

More ablation studies will be shown in Appendix E.

Table 4: Ablation studies ($\times 10^{-3}$) on MD17 dataset. Results averaged across 3 runs.

|  | Aspirin | Benzene | Ethanol | Malonaldehyde | Naphthalene | Salicylic | Toluene | Uracil |
|---|---|---|---|---|---|---|---|---|
| ESTAG | **0.063** | **0.003** | **0.099** | **0.101** | **0.068** | **0.047** | **0.079** | 0.066 |
| w/o EDFT | 0.079 | **0.003** | 0.108 | 0.148 | 0.104 | 0.145 | 0.102 | **0.063** |
| w/o Attention | 0.087 | 0.004 | 0.104 | 0.112 | 0.129 | 0.095 | 0.097 | 0.078 |
| w/o Equivariance | 0.762 | 0.114 | 0.458 | 0.604 | 0.738 | 0.698 | 0.690 | 0.680 |
| w/o Temporal | 0.084 | **0.003** | 0.111 | 0.139 | 0.141 | 0.098 | 0.153 | 0.071 |

Table 5: MSE on Ethanol $w.r.t.$ the number of layers $L$.

| $L$ | 1 | 2 | 3 | 4 | 5 | 6 |
|---|---|---|---|---|---|---|
| MSE ($\times 10^{-4}$) | 1.25 | 0.990 | 1.096 | 1.022 | 1.042 | 1.028 |

## 6 Conclusion

In this paper, we propose ESTAG, an end-to-end equivariant architecture for physical dynamics modeling. ESTAG first extracts frequency features via a novel Equivariant Discrete Fourier Transform (EDFT), and then leverages Equivariant Spatial Module (ESM) and an attentive Equivariant Temporal Module (ETM) to refine the coordinate in space and time domain alternatively. Comprehensive experiments over multiple tasks verify the superiority of ESTAG from molecular-level, protein-level, and to macro-level. Necessary ablations, visualizations, and analyses are also provided to support the validity of our design as well as the generalization of our method. One potential limitation of our model is that we only enforce the E(3) symmetry while other inductive bias like the energy conservation law is also required in physical scenarios.

In the future, we will continue extending our benchmark with more tasks and datasets and evaluate more baselines to validate the effectiveness of our model. It is also promising to extend our model to multi-scale GNN (like SGNN [17], REMuS-GNN [27], BSMS-GNN [4] and MS-MGN [10]), which is useful particularly for industrial-level applications involving huge graphs. Besides, it is valuable to employ our simulation method as a basic block for other applications such as drug discovery, material design, robotic control, etc.

## 7 Acknowledgement

This work was jointly supported by the following projects: the Scientific Innovation 2030 Major Project for New Generation of AI under Grant NO. 2020AAA0107300, Ministry of Science and Technology of the People's Republic of China; the National Natural Science Foundation of China (62006137); Beijing Nova Program (20230484278); the Fundamental Research Funds for the Central Universities, and the Research Funds of Renmin University of China (23XNKJ19); Tencent AI Lab Rhino-Bird Focused Research Program (RBFR2023005); Ant Group through CCF-Ant Research Fund (CCF-AFSG RF20220204); Public Computing Cloud, Renmin University of China.

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

# A  Proof of ESTAG's Equivariance

**Theorem A.1.** *We denote ESTAG as $\vec{\boldsymbol{X}}(T) = \phi\left(\{(\boldsymbol{H}(t), g \cdot \vec{\boldsymbol{X}}(t), \boldsymbol{A})\}_{t=0}^{T-1}\right)$, then $\phi$ is E(3)-equivariant.*

*Proof.* **1.** We firstly prove that EDFT is E(3)-equivariant.

$$\boldsymbol{O}\vec{\boldsymbol{f}}_i(k) = \sum_{t=0}^{T-1} e^{-i'\frac{2\pi}{T}kt} \left(\boldsymbol{O}\vec{\boldsymbol{x}}_i(t) + \boldsymbol{b} - \overline{\boldsymbol{O}\vec{\boldsymbol{x}}(t) + \boldsymbol{b}}\right),$$

$$\boldsymbol{A}_{ij}(k) = w_k(\boldsymbol{h}_i)w_k(\boldsymbol{h}_j)|\langle \boldsymbol{O}\vec{\boldsymbol{f}}_i(k), \boldsymbol{O}\vec{\boldsymbol{f}}_j(k)\rangle|,$$

$$\boldsymbol{c}_i(k) = w_k(\boldsymbol{h}_i)\|\boldsymbol{O}\vec{\boldsymbol{f}}_i(k)\|^2.$$

**2.** We secondly prove the E(3)-equivariance of ESM.

$$\boldsymbol{m}_{ij} = \phi_m\left(\boldsymbol{h}_i^{(l)}(t), \boldsymbol{h}_j^{(l)}(t), \|\boldsymbol{O}\vec{\boldsymbol{x}}_{ij}^{(l)}(t)\|^2, \boldsymbol{A}_{ij}\right),$$

$$\boldsymbol{h}_i^{(l+1)}(t) = \phi_h\left(\boldsymbol{h}_i^{(l)}(t), \boldsymbol{c}_i(k), \sum_{j\neq i}\boldsymbol{m}_{ij}\right),$$

$$\boldsymbol{O}\vec{\boldsymbol{a}}_i(t) = \frac{1}{|\mathcal{N}(i)|}\sum_{j\in\mathcal{N}(i)}\boldsymbol{O}\vec{\boldsymbol{x}}_{ij}^{(l)}(t)\phi_x(\boldsymbol{m}_{ij}),$$

$$\boldsymbol{O}\vec{\boldsymbol{x}}_i^{(l+1)}(t) + \boldsymbol{b} = \boldsymbol{O}\vec{\boldsymbol{x}}_i^{(l)}(t) + \boldsymbol{b} + \boldsymbol{O}\vec{\boldsymbol{a}}_i^{(l+1)}(t).$$

**3.** We then prove that ETM is E(3)-equivariant.

$$\boldsymbol{q}_i^{(l)}(t) = \phi_q\left(\boldsymbol{h}_i^{(l)}(t)\right),$$

$$\boldsymbol{k}_i^{(l)}(t) = \phi_k\left(\boldsymbol{h}_i^{(l)}(t)\right),$$

$$\boldsymbol{v}_i^{(l)}(t) = \phi_v\left(\boldsymbol{h}_i^{(l)}(t)\right),$$

$$\alpha_i^{(l)}(ts) = \frac{\exp(\boldsymbol{q}_i^{(l)}(t)^\top \boldsymbol{k}_i^{(l)}(s))}{\sum_{s=0}^{t}\exp(\boldsymbol{q}_i^{(l)}(t)^\top \boldsymbol{k}_i^{(l)}(s))},$$

$$\boldsymbol{h}_i^{(l+1)}(t) = \boldsymbol{h}_i^{(l)}(t) + \sum_{s=0}^{t}\alpha_i^{(l)}(ts)\boldsymbol{v}_i^{(l)}(s),,$$

$$\boldsymbol{O}\vec{\boldsymbol{x}}_i^{(l+1)}(t) + \boldsymbol{b} = \boldsymbol{O}\vec{\boldsymbol{x}}_i^{(l)}(t) + \boldsymbol{b} + \sum_{s=0}^{t}\alpha_i^{(l)}(ts)\,\boldsymbol{O}\vec{\boldsymbol{x}}_i^{(l)}(ts)\phi_x(\boldsymbol{v}_i^{(l)}(s)).$$

**4.** We finally prove that the linear pooling is equivariant:

$$\boldsymbol{O}\vec{\boldsymbol{x}}_i^*(T) + \boldsymbol{b} = \boldsymbol{O}\hat{\boldsymbol{X}}_i\boldsymbol{w} + \boldsymbol{O}\vec{\boldsymbol{x}}_i^{(L)}(T-1) + \boldsymbol{b}.$$

$\square$

# B  Full Algorithm Details

In the main body of the paper, for better readability, we present the implementation details of ESTAG. Here, we combine them into one singe algorithmic flowchart in Algorithm 1.

# C Extended Models

## C.1 Multi-channel ESTAG

For proteins, there are four backbone atoms ( N, $C_\alpha$, C, O) in residue $i$, hence the above mentioned node position vector $\boldsymbol{x}_i(t) \in \mathbb{R}^3$ is extended to a 4-channel position matrix $\boldsymbol{X}_i(t) \in \mathbb{R}^{3 \times 4}$. Particularly, we denote $\vec{\boldsymbol{x}}_i^\alpha(t)$, a certain column from $\boldsymbol{X}_i(t)$ as the position of $C_\alpha$ at time $t$.

**EDFT:**

$$\vec{\boldsymbol{f}}_i(k) = \sum_{t=0}^{T-1} e^{-i' \frac{2\pi}{T} kt} \left( \vec{\boldsymbol{x}}_i^\alpha(t) - \overline{\vec{\boldsymbol{x}}^\alpha(t)} \right),$$

$$\boldsymbol{A}_{ij}(k) = w_k(\boldsymbol{h}_i) w_k(\boldsymbol{h}_j) |\langle \vec{\boldsymbol{f}}_i(k), \vec{\boldsymbol{f}}_j(k) \rangle|,$$

$$\boldsymbol{c}_i(k) = w_k(\boldsymbol{h}_i) \|\vec{\boldsymbol{f}}_i(k)\|^2.$$

**ESM:**

$$\boldsymbol{m}_{ij} = \phi_m \left( \boldsymbol{h}_i^{(l)}(t), \boldsymbol{h}_j^{(l)}(t), \frac{(\vec{\boldsymbol{X}}_{ij}^{(l)}(t))^\top \vec{\boldsymbol{X}}_{ij}^{(l)}(t)}{\|(\vec{\boldsymbol{X}}_{ij}^{(l)}(t))^\top \vec{\boldsymbol{X}}_{ij}^{(l)}(t)\|_F}, \boldsymbol{A}_{ij} \right),$$

$$\boldsymbol{h}_i^{(l+1)}(t) = \phi_h \left( \boldsymbol{h}_i^{(l)}(t), \boldsymbol{c}_i(k), \sum_{j \neq i} \boldsymbol{m}_{ij} \right),$$

$$\vec{\boldsymbol{A}}_i^{(l)}(t) = \frac{1}{|\mathcal{N}(i)|} \sum_{j \in \mathcal{N}(i)} \vec{\boldsymbol{X}}_{ij}^{(l)}(t) \phi_{\boldsymbol{X}}(\boldsymbol{m}_{ij}),$$

$$\vec{\boldsymbol{X}}_i^{(l+1)}(t) = \vec{\boldsymbol{X}}_i^{(l)}(t) + \vec{\boldsymbol{A}}_i^{(l)}(t).$$

**ETM:**

$$\alpha_i^{(l)}(ts) = \frac{\exp(\boldsymbol{q}_i^{(l)}(t)^\top \boldsymbol{k}_i^{(l)}(s))}{\sum_{s=0}^{t} \exp(\boldsymbol{q}_i^{(l)}(t)^\top \boldsymbol{k}_i^{(l)}(s))},$$

$$\boldsymbol{h}_i^{(l+1)}(t) = \boldsymbol{h}_i^{(l)}(t) + \sum_{s=0}^{t} \alpha_i^{(l)}(ts) \boldsymbol{v}_i^{(l)}(s),$$

$$\vec{\boldsymbol{X}}_i^{(l+1)}(t) = \vec{\boldsymbol{X}}_i^{(l)}(t) + \sum_{s=0}^{t} \alpha_i^{(l)}(ts) \vec{\boldsymbol{X}}_i^{(l)}(ts) \phi_{\boldsymbol{X}}(\boldsymbol{v}_i^{(l)}(s)),$$

where

$$\boldsymbol{q}_i^{(l)}(t) = \phi_q \left( \boldsymbol{h}_i^{(l)}(t) \right),$$

$$\boldsymbol{k}_i^{(l)}(t) = \phi_k \left( \boldsymbol{h}_i^{(l)}(t) \right),$$

$$\boldsymbol{v}_i^{(l)}(t) = \phi_v \left( \boldsymbol{h}_i^{(l)}(t) \right).$$

# D More Experimental Details and Results

## D.1 More Details on MD17

**Experiment setup and hyper-parameters.** We use the following hyper-parameters across all experimental evaluations: batch size 100, the number of epochs 500, weight decay $1 \times 10^{-12}$, the number of layers 4 (we consider one ESTAG includes two layers, i.e. ESM and ETM), hidden dim 16, Adam optimizer with learning rate $5 \times 10^{-3}$. We set the length of previous time series $L = 10$ and the interval between two timestamps $\Delta t = 10$. The number of training, validation and testing sets are 500, 2000 and 2000, respectively.

## D.2 More Details on Protein

**Experiment setup and hyper-parameters.** We use the following hyper-parameters across all experimental evaluations: batch size 100, the number of epochs 500, weight decay $1 \times 10^{-12}$, the number of layers 4 (we consider one ESTAG includes two layers, i.e. ESM and ETM), hidden dim 16, Adam optimizer with learning rate $5 \times 10^{-5}$. We divide the whole dataset into training, validation and testing sets by a ratio of 6:2:2, resulting in the numbers of the three sets are 2482, 827 and 827 respectively. The number of previous timestamps is $T = 10$ and the interval between timestamps is $\Delta t = 5$ frames.

## D.3 More Details on Motion

**Experiment setup and hyper-parameters.** We use the following hyper-parameters across all experimental evaluations: batch size 100, the number of epochs 500, weight decay $1 \times 10^{-12}$, the number of layers 4 (we consider one ESTAG includes two layers, i.e. ESM and ETM), hidden dim 16, Adam optimizer with learning rate $5 \times 10^{-3}$. We set the length of previous time series $L = 10$ and the interval between two timestamps $\Delta t = 5$. The training/validation/testing sets sizes are 3000/800/800 respectively.

## D.4 Long-term recurrent forecasting

We additionally explore the performance of the proposed method in long-term recurrent forecasting. The setting in our current experiment predicts only one frame at a time . Here we recurrently predict the future frames at time $T, T + \Delta T, T + 2\Delta T, \cdots, T + 10\Delta T$ (the value of $\Delta T$ follows the setting in the Section 5) in a rollout manner, where the currently-predicted frame will be used as the input for the next frame prediction, within a sliding window of length $T$. Note that the recurrent forecasting task is more challenging than the original scenario, and we need to make some extra improvements to prevent accumulated errors over time. Particularly for our method, we change the forward attention mechanism to be full attention mechanism (namely replacing $t$ with $T - 1$ in the superscript of the summation of Eq.10 and Eq.12 10), as we observe that under the recurrent setting, forward attention tends to lead to biased predictions. The results are reported in Figure 5, where we verify that the rollout version of ESTAG delivers generally smaller MSE than all compared methods for all time steps.

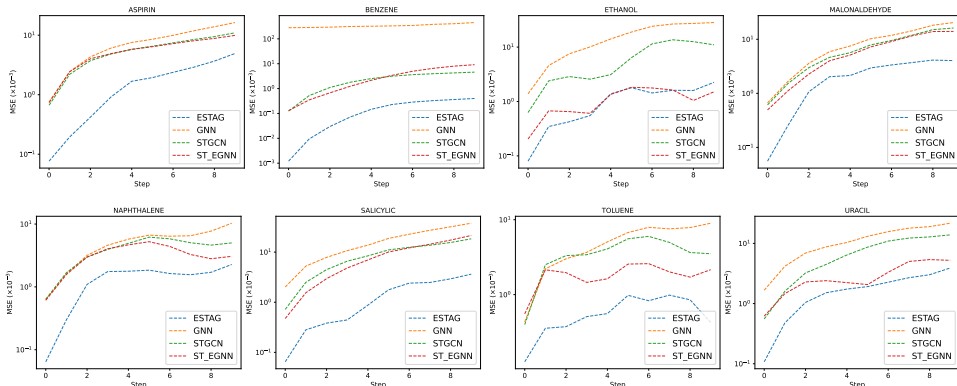

Figure 5: The rollout-MSE curves on 8 molecules in MD17. Our model generally achieves the lowest MSE.

## D.5 More visualization

**Visualization of data.** To validate that the movement of MD17 molecules is periodical, in Figure 6 we depict the trajectory of one randomly selected atom in each molecule, from timestep 127947 to timestep 327947. It is obvious that almost all molecules move with period.

**Visualization of model parameters.** Moreover, we display the attention map in ETM of MAL-ONALDEHYDE's atom O and temporal pooling weights of all molecules, as shown in Figure 7.

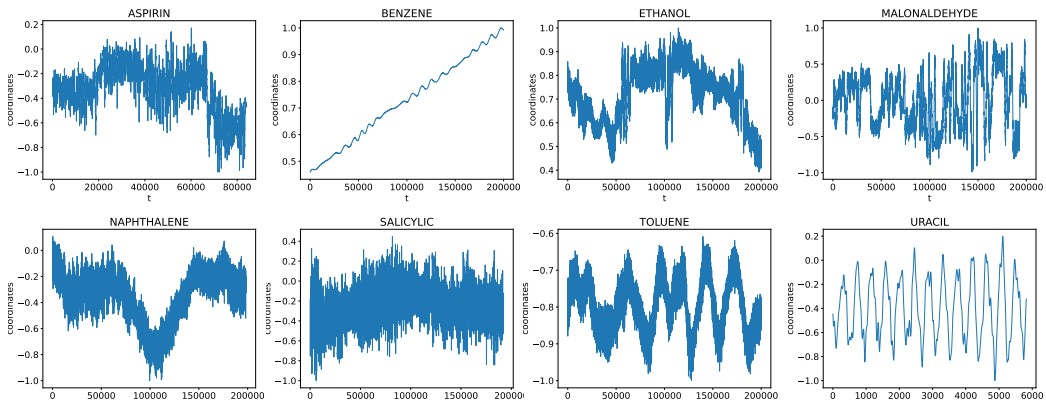

Figure 6: The movement trajectory of 8 molecules in MD17.

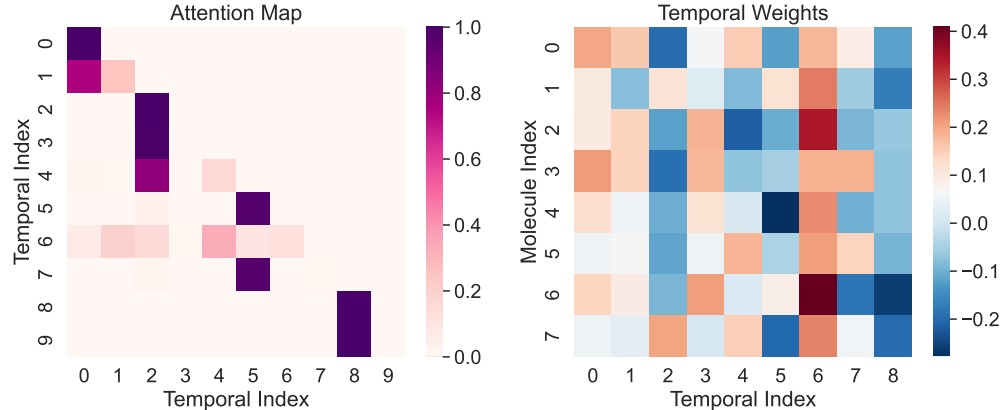

Figure 7: The visualization of temporal attention Map (left) and temporal pooling weights (right).

**Visualization of the prediction results on MD17 dataset.** For those non-obvious cases in Figure 4, the unclarity is mainly caused by 2D printing that is hard to depict 3D conformations such as angle deviation. For better visualization, we choose the molecule URACIL which is not clearly visualized, and render its 3D atom coordinates in Figure 8. ESTAG yields more accurate 3D prediction than ST_EGNN, which is consistent with the MSE difference.

**Visualization of the prediction results on Protein dataset.** We further evaluate the predicted protein by visualizing it alongside the ground truth in Figure 9, using the ChimeraX software [28]. It is obvious that protein predicted by ESTAG aligns closer to the ground truth and ST_EGNN fails to predict the alpha helix which we highlight in arrow.

**Visualization of the prediction results on Motion dataset.** Following the visualization manner used for molecules, we represent the entities from both walking and basketball motion scenarios within a 3D coordinate system, in Figure 10 and Figure 11 respectively. It confirms the superiority of ESTAG, particularly in the significantly more challenging task of simulating basketball motion.

# E  More Ablation Studies

## E.1  The impact of the number of past timestamps $T$

We investigate effect of the number of previous timestamps $T$ on the $Ethanol$ dataset. We vary $T$ from 3, 5, 8, 10, 20, 30, 40, 50, 60, 70 and present the results in Table 6 and Figure 12. To balance the trade-off between efficiency and performance, we choose T=10 in our experiments.

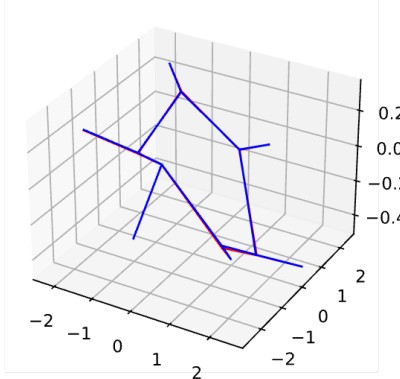 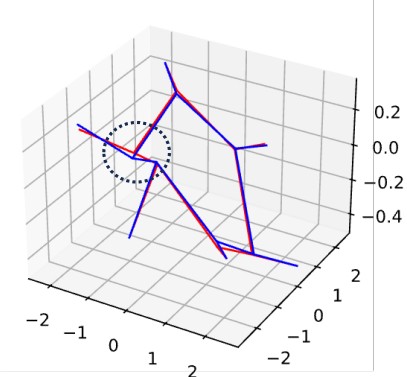

Figure 8: 3D visualization of URACIL for comparison between ESTAG (left, MSE=$5.983 \times 10^{-5}$) and ST_EGNN (right, MSE=$6.393 \times 10^{-4}$). The ground truths are in red while the predicted states are in blue. The molecule predicted by ESTAG is better aligned with the ground truth, particularly for the regions highlighted by circles.

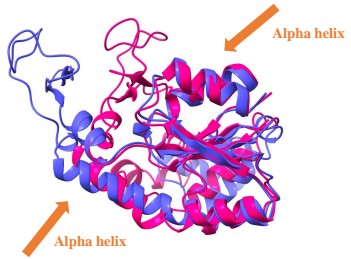 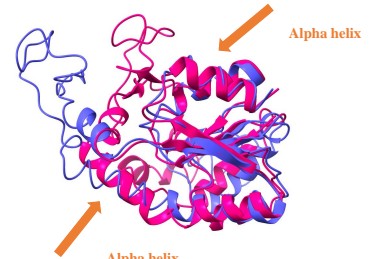

Figure 9: Comparison on Protein dataset between ESTAG (left, MSE=1.014) and ST_EGNN (right, MSE=1.388). The ground truths are in red while the predicted states are in blue.

## E.2 The impact of the learnable parameter $w_k$ in EDFT module

In section 4.1, We have noticed that $w_k$ acts like spectral filters of the $k$-th frequency and enables us to select related frequency for the prediction. Here, we conduct ablation study on MD17 dataset to analyse how significant the parameter is. From the results in Table 7, $w_k$ is indeed capable of improving the prediction of physical dynamics.

## E.3 Ablation studies on Protein and Motion datasets

Here we conduct several ablations on Protein and Motion dataset similar to MD17 and show the results in Table 8 and Table 9, respectively.

Table 6: MSE on Ethanol $w.r.t.$ the number of previous timestamps $T$

| $T$ | 3 | 5 | 8 | 10 | 20 | 30 | 40 | 50 | 60 | 70 |
|---|---|---|---|---|---|---|---|---|---|---|
| MSE ($\times 10^{-4}$) | 1.760 | 1.395 | 1.064 | 0.990 | 0.924 | 1.017 | 0.962 | 1.009 | 0.992 | 1.019 |

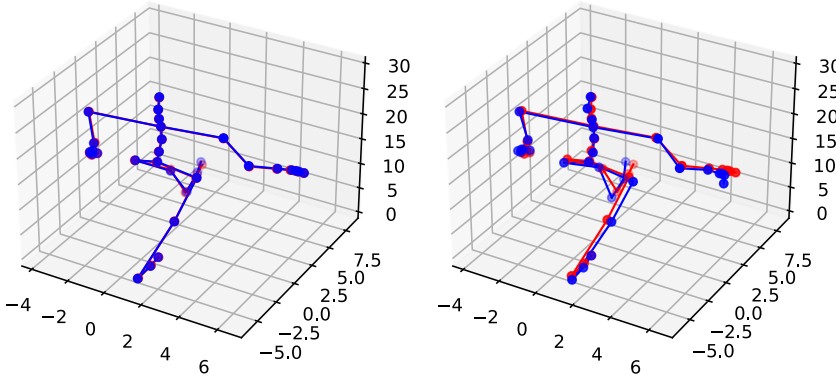

Figure 10: Comparison on Motion walk subject between ESTAG (left, MSE=0.0048) and ST_EGNN (right, MSE=0.0811). The ground truths are in red while the predicted states are in blue.

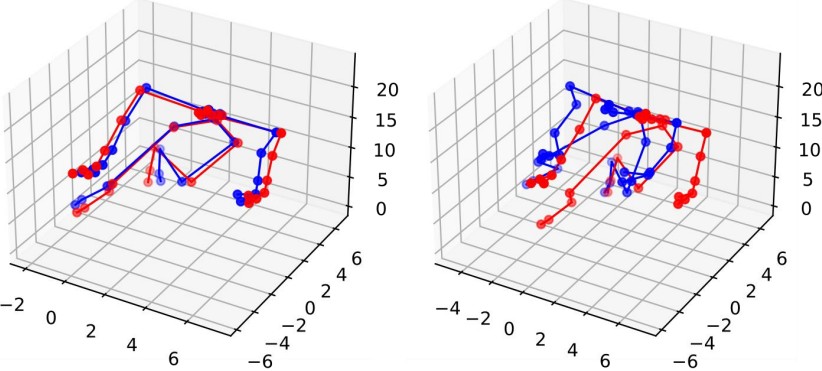

Figure 11: Comparison on Motion basketball subject between ESTAG (left, MSE=0.0749) and ST_EGNN (right, MSE=2.6380). The ground truths are in red while the predicted states are in blue.

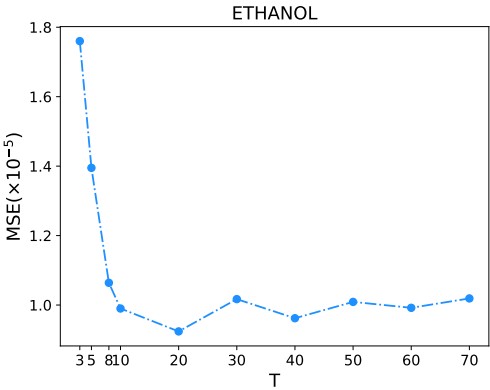

Figure 12: MSE on Ethanol $w.r.t.$ the number of previous timestamps $T$

Table 7: Ablation studies ($\times 10^{-3}$) of learnable parameter $w_k$ on MD17 dataset. Results averaged across 3 runs.

|              | Aspirin | Benzene | Ethanol | Malonaldehyde | Naphthalene | Salicylic | Toluene | Uracil |
| ------------ | ------- | ------- | ------- | ------------- | ----------- | --------- | ------- | ------ |
| ESTAG        | **0.063** | **0.003** | **0.099** | **0.101**     | **0.068**   | **0.047** | **0.079** | **0.066** |
| w/o $w_k$    | 0.071   | **0.003** | 0.102   | 0.104         | 0.081       | 0.081     | **0.079** | 0.069  |

Table 8: Ablation study on Protein dataset.   Table 9: Ablation study on Motion dataset.

| Model | MSE |
|---|---|
| ESTAG | **1.471** |
| w/o EDFT | 1.490 |
| w/o Attention | 1.493 |
| w/o Equivariance | 2.011 |
| w/o Temporal | 1.472 |

| Model | MSE ($\times 10^{-1}$) |
|---|---|
| ESTAG | **0.040** |
| w/o EDFT | 0.053 |
| w/o Attention | 0.047 |
| w/o Equivariance | 0.702 |
| w/o Temporal | 0.044 |

# F Codes

The codes of ESTAG are available at: https://github.com/ManlioWu/ESTAG.

---

**Algorithm 1** Equivariant Spatio-Temporal Attentive Graph Networks (ESTAG)

---

**Input:** Initial historical graph series $\{(\boldsymbol{H}^{(0)}(t), \vec{\boldsymbol{X}}^{(0)}(t), \boldsymbol{A}\}_{t=0}^{T-1}$.

**for** $i = 1$ **to** $N$ **do**

    Equivariant Discrete Fourier Transform (EDFT):

$$\vec{\boldsymbol{f}}_i(k) = \sum_{t=0}^{T-1} e^{-i' \frac{2\pi}{T} kt} \left( \vec{\boldsymbol{x}}_i(t) - \overline{\vec{\boldsymbol{x}}(t)} \right), \tag{15}$$

$$\boldsymbol{A}_{ij}(k) = w_k(\boldsymbol{h}_i) w_k(\boldsymbol{h}_j) |\langle \vec{\boldsymbol{f}}_i(k), \vec{\boldsymbol{f}}_j(k) \rangle|, \tag{16}$$

$$\boldsymbol{c}_i(k) = w_k(\boldsymbol{h}_i) \|\vec{\boldsymbol{f}}_i(k)\|^2. \tag{17}$$

**end for**

**for** $l = 1$ **to** $L$ **do**

  **for** $t = 0$ **to** $T - 1$ **do**

    Equivariant Spatial Module (ESM):

$$\boldsymbol{m}_{ij} = \phi_m \left( \boldsymbol{h}_i^{(l-1)}(t), \boldsymbol{h}_j^{(l-1)}(t), \|\vec{\boldsymbol{x}}_{ij}^{(l-1)}(t)\|^2, \boldsymbol{A}_{ij} \right), \tag{18}$$

$$\boldsymbol{h}_i^{(l-0.5)}(t) = \phi_h \left( \boldsymbol{h}_i^{(l-1)}(t), \boldsymbol{c}_i(k), \sum_{j \neq i} \boldsymbol{m}_{ij} \right), \tag{19}$$

$$\vec{\boldsymbol{a}}_i^{(l-0.5)}(t) = \frac{1}{|\mathcal{N}(i)|} \sum_{j \in \mathcal{N}(i)} \vec{\boldsymbol{x}}_{ij}^{(l-0.5)}(t) \phi_x(\boldsymbol{m}_{ij}), \tag{20}$$

$$\vec{\boldsymbol{x}}_i^{(l-0.5)}(t) = \vec{\boldsymbol{x}}_i^{(l-1)}(t) + \vec{\boldsymbol{a}}_i^{(l-0.5)}(t). \tag{21}$$

  **end for**

  **for** $i = 1$ **to** $N$ **do**

    Equivariant Temporal Module (ETM)

$$\alpha_i^{(l-0.5)}(ts) = \frac{\exp(\boldsymbol{q}_i^{(l-0.5)}(t)^\top \boldsymbol{k}_i^{(l-0.5)}(s))}{\sum_{s=0}^{t} \exp(\boldsymbol{q}_i^{(l-0.5)}(t)^\top \boldsymbol{k}_i^{(l-0.5)}(s))}, \tag{22}$$

$$\boldsymbol{h}_i^{(l)}(t) = \boldsymbol{h}_i^{(l-0.5)}(t) + \sum_{s=0}^{t} \alpha_i^{(l-0.5)}(ts) \boldsymbol{v}_i^{(l-0.5)}(s), \tag{23}$$

$$\vec{\boldsymbol{x}}_i^{(l)}(t) = \vec{\boldsymbol{x}}_i^{(l-0.5)}(t) + \sum_{s=0}^{t} \alpha_i^{(l-0.5)}(ts) \vec{\boldsymbol{x}}_i^{(l-0.5)}(ts) \phi_x(\boldsymbol{v}_i^{(l-0.5)}(s)), \tag{24}$$

    where

$$\boldsymbol{q}_i^{(l-0.5)}(t) = \phi_q \left( \boldsymbol{h}_i^{(l-0.5)}(t) \right), \tag{25}$$

$$\boldsymbol{k}_i^{(l-0.5)}(t) = \phi_k \left( \boldsymbol{h}_i^{(l-0.5)}(t) \right), \tag{26}$$

$$\boldsymbol{v}_i^{(l-0.5)}(t) = \phi_v \left( \boldsymbol{h}_i^{(l-0.5)}(t) \right). \tag{27}$$

  **end for**

**end for**

Equivariant linear pooling:

$$\vec{\boldsymbol{x}}_i^*(T) = \hat{\boldsymbol{X}}_i \boldsymbol{w} + \vec{\boldsymbol{x}}_i^{(L)}(T-1), \tag{28}$$

where $\hat{\boldsymbol{X}}_i = [\vec{\boldsymbol{x}}_i^{(L)}(0) - \vec{\boldsymbol{x}}_i^{(L)}(T-1), \vec{\boldsymbol{x}}_i^{(L)}(1) - \vec{\boldsymbol{x}}_i^{(L)}(T-1), \cdots, \vec{\boldsymbol{x}}_i^{(L)}(T-2) - \vec{\boldsymbol{x}}_i^{(L)}(T-1)]$.

**Output:** $\vec{\boldsymbol{X}}^*(T)$

---

