# OpenReview forum: "Equivariant Spatio-Temporal Attentive Graph Networks to Simulate Physical Dynamics"
_NeurIPS.cc/2023/Conference — NeurIPS 2023 poster_

### Official Review · Reviewer_TWyG · 2023-07-05

**Soundness:** 4 excellent
**Presentation:** 3 good
**Contribution:** 3 good
**Rating:** 7
**Confidence:** 5

**Summary:**

This paper introduces an E(3)-invariant temporal attention scheme, calculated with the help of discrete Fourier transform, within the E(3)-equivariant GNN framework. The overall idea of considering higher-order temporal effects in physics is sound, and the formulation appears to be correct.

There are a few typos that do not affect the overall scoring, I would recommend the authors do a full proofreading.

There may be missing references and potentially a missing benchmark to compare with. I recommend adding these (in **Weaknesses**).

Overall, this work is solid, and I recommend accepting it for Neural IPS 2023.

**Strengths:**

The idea presented in the paper is novel, although not groundbreaking. It fills a gap in the existing framework and the direction is practical and meaningful.

There are empirical improvements.

The illustrations are very easy to follow.

**Weaknesses:**

There are some typos in the paper, such as the missing year in reference [16]. I recommend thorough proofreading.

—Lack of previous SOTA for comparison—

[1] Chen, Runfa and Han, Jiaqi and Sun, Fuchun and Huang, Wenbing. "Subequivariant Graph Reinforcement Learning in 3D Environments". Link: https://arxiv.org/abs/2305.18951

—Lack of reference for future improvements—

One future direction I have, which has already been used in [1], is combining equivariance with multi-scale (MS) GNN, as most industrial-level applications involve huge graphs. Therefore, the following papers should be cited as future works. Note that [2] also combines equivariance with MS, similar to [1]. However, since there are significant differences in graph type and application needs between this paper and [2], it is not suggested to compare them directly (but they should still be cited).

[2] Lino, Mario and Fotiadis, Stathi and Bharath, Anil A and Cantwell, Chris D. “Multi-scale rotation-equivariant graph neural networks for unsteady Eulerian fluid dynamics”. Link: https://pubs.aip.org/aip/pof/article/34/8/087110/2847850

[3] Cao, Yadi, Menglei Chai, Minchen Li, and Chenfanfu Jiang. "Efficient learning of mesh-based physical simulation with bi-stride multi-scale graph neural network.". Link: https://openreview.net/forum?id=2Mbo7IEtZW

[4] Meire Fortunato, Tobias Pfaff, Peter Wirnsberger, Alexander Pritzel, Peter Battaglia. “MultiScale MeshGraphNets”. Link: https://arxiv.org/abs/2210.00612

**Questions:**

Please refer to the **Weaknesses** section.

**Limitations:**

Please see the comments regarding "combining equivariance with multi-scale" in the **Weaknesses** section. The attention module for higher-order temporal relationships will significantly increase complexity, which may limit the application in industrial scenarios with huge graphs. The author should either acknowledge this limit, and/or analyze potential remedies for this overhead.

---

> ### Author Rebuttal · Authors · 2023-08-10
>
> Thanks! Your feedback is instrumental in strengthening our paper.
>
> >Q1: There are some typos in the paper, such as the missing year in reference [16]. I recommend thorough proofreading.
>
> Thank you very much for the mentioned typos, and we will fix them and proofread our paper carefully.
>
> >Q2: —Lack of previous SOTA for comparison—
>
> Thank you for raising the related paper [A], which will be cited in the revised paper. Notably, the reference [A] clearly differs from our paper in two aspects:  Firstly, [A] mainly incorporates equivariance into Reinforcement Learning (RL) for morphology-agnostic locomotion learning, whereas our paper aims at equivariant dynamics simulation. Secondly, both the policy and Q functions used in [A] still belong to the frame-to-frame prediction paradigm and are constructed under the Markovian assumption, while our model is of the spatio-temporal form to pursue non-Markovian modeling. We will add the above discussions to the revised paper.
>
> >Q3: —Lack of reference for future improvements—
>
> Nice suggestion! We agree that combining equivariance with multi-scale (MS) GNN is valuable particuarly for  industrial-level applications involve huge graphs. We are willing to cite and discuss the mentioned papers [B-D], and consider equivariant MS GNN as a future exploration direction.
>
> >Q4: On limitations.
>
> Thanks for the suggestion. We will discuss the efficiency issue for industrial scenarios with huge graphs. As suggested by the reviewer, exploring multi-scale architectures upon our model is potential to reduce the complexity overhead, which will be acknowledged in the revised paper.
>
>
> [A] Chen, Runfa and Han, Jiaqi and Sun, Fuchun and Huang, Wenbing. "Subequivariant Graph Reinforcement Learning in 3D Environments".
>
> [B] Lino, Mario and Fotiadis, Stathi and Bharath, Anil A and Cantwell, Chris D. “Multi-scale rotation-equivariant graph neural networks for unsteady Eulerian fluid dynamics”.
>
> [C] Cao, Yadi, Menglei Chai, Minchen Li, and Chenfanfu Jiang. "Efficient learning of mesh-based physical simulation with bi-stride multi-scale graph neural network.".
>
> [D] Meire Fortunato, Tobias Pfaff, Peter Wirnsberger, Alexander Pritzel, Peter Battaglia. “MultiScale MeshGraphNets”.

---

> > ### Comment · Reviewer_TWyG · 2023-08-10
> > **Reply**
> >
> > Thanks for your reply. I made a mistake in suggesting [A] to you. It should have been
> >
> > Learning Physical Dynamics with Subequivariant Graph Neural Networks (NeurIPS 22), https://arxiv.org/pdf/2210.06876.pdf
> >
> > Considering the time is not enough for you to add any comparison before the discussion period. Now I can only suggest that the correct version be cited now. If you got accepted in the end, you should try comparing to this one in the final revision.
> >
> > Best,

---

> > > ### Author Response · Authors · 2023-08-11
> > > **Thank you for your feedback**
> > >
> > > Thank you for your prompt response! Your feedback is very valuable. The suggested paper effectively utilizes hierarchy and multi-scale techniques in the analysis of large-scale graphs, which can further enhance our research. We will make sure to cite and compare it in the final version of our paper.

---

### Official Review · Reviewer_pTt6 · 2023-07-05

**Soundness:** 3 good
**Presentation:** 3 good
**Contribution:** 3 good
**Rating:** 6
**Confidence:** 3

**Summary:**

This paper addresses the Markov limitation of previous methods in simulating physical dynamics by treating it as a spatio-temporal prediction task. The authors propose Equivariant Graph Neural Networks (GNNs) to account for the non-Markovian nature of the systems. Additionally, they design three components to extract spatio-temporal features while preserving equivariance. The experiments conducted on three real datasets demonstrate that the proposed method surpasses previous approaches and validate the effectiveness of the three designed components.

**Strengths:**

1. Identification of Markov limitation: The paper recognizes the Markov limitation present in previous methods and appropriately considers the non-Markovian nature of the systems. This approach is well-founded.
2. Equivariant property preservation: The designed components successfully maintain the equivariant property while extracting spatio-temporal features. The paper also provides theoretical evidence of the proposed Equivariant Spatio-Temporal Attentive Graph (ESTAG) being E(3)-equivariant.
3. Comprehensive experimental validation: The paper includes a substantial number of experiments to substantiate the proposed methods. The results consistently demonstrate the superiority of the proposed approach over alternative methods and confirm the effectiveness of the designed Equivariant Discrete Fourie rTransform (EDFT), Equivariant Spatial Module (ESM), and Equivariant Temporal Module (ETM).

**Weaknesses:**

1. Lack of clarity regarding EDFT: The paper does not provide sufficient explanation of how EDFT improves prediction accuracy. It is important to clarify the underlying mechanisms and intuition behind the proposed component.
2. Need for additional experiments: It would be valuable to conduct further experiments to explore the performance of the proposed method in long-term recurrent forecasting. Providing results in such scenarios would enhance the understanding of the model's capabilities.

**Questions:**

Please refer to weaknesses.

**Limitations:**

The authors do not address the limitations. Please refer to weaknesses.

---

> ### Author Rebuttal · Authors · 2023-08-10
>
> We are grateful for your positive and constructive comments, and provide the answers to your questions below.
>
> >Q1: Lack of clarity regarding EDFT: The paper does not provide sufficient explanation of how EDFT improves prediction accuracy. It is important to clarify the underlying mechanisms and intuition behind the proposed component.
>
> Sorry for the insufficient clarity. Our use of EDFT is inspired by the observation of molecular trajectories on MD17. As already visualized in Figure 1 in the supplementary material, the molecular dynamics exhibit certain periodicity in terms of different frequencies. This motivates us to first transform the trajectories from time domain to frequency domain, and then compute the inner-product between different frequencies in Eq.4. In signal processing, a well-known theorem is that the Fourier transform of cross-correlation between two signals is equal to the inner-product between the Fourier transform of two signals. In this sense, Eq.4 is able to measure the cross-correlation (thus similarity) between any two trajectories in the frequency domain, and thus can be regarded as the adjacent value between nodes in the message passing in Eq. 6. Similarly, Eq. 5 computes the amplitude which is regarded as a node feature in the message passing in Eq.7. We will include the above explanations into the revised version.
>
> >Q2: Need for additional experiments: It would be valuable to conduct further experiments to explore the performance of the proposed method in long-term recurrent forecasting. Providing results in such scenarios would enhance the understanding of the model's capabilities.
>
> Nice suggestion!  We additionally explore the performance of the proposed method in long-term recurrent forecasting as suggested. The setting in our current paper predicts only one frame at time $T$. Here we recurrently predict the future frames at time $T, T+\Delta t, T+2\Delta t, \cdots, T+10\Delta t $ (the value of $\Delta t$ follows the setting in the paper) in a rollout manner, where the currently-predicted frame will be used as the input for the prediction of the next frame, within a sliding window of length $T$. Note that the recurrent forecasting task is more challenging than the original scenario, and we need to make some extra improvement to prevent accumulated errors over time. Particularly for our method, we change the forward attention mechanism to be full attention mechanism (namely replacing $t$ with $T-1$ in the superscript of the summation of Eq.10 and Eq.12), as we find that the foward attention is prone to biased prediction under the recurrent setting.
> The results are reported in Figure A5 (General Response), where we verify that the rollout version of ESTAG delivers generally smaller MSE than all compared methods for all time steps.  We will contain the evaluation of the long-term recurrent forecasting in the revised paper.

---

> > ### Comment · Reviewer_pTt6 · 2023-08-18
> >
> > Thank you for your response. I value both the quality of the paper and the thoroughness of the rebuttal. Consequently, I have chosen to maintain the positive score.

---

> > > ### Author Response · Authors · 2023-08-18
> > > **Thanks for your reply**
> > >
> > > Dear reviewer:
> > >
> > > Thank you for taking the time to respond. We greatly appreciate your endorsement of our work.

---

### Official Review · Reviewer_Gpx3 · 2023-07-06

**Soundness:** 2 fair
**Presentation:** 3 good
**Contribution:** 3 good
**Rating:** 6
**Confidence:** 4

**Summary:**

This paper studies the non-Markovian dynamics that often appear in physical systems and proposes a spatio-temporal E(3) equivariant graph network that moves beyond the simple frame-to-frame prediction task.
The authors introduce an equivariant feature extraction method based on Fourier Transform, as well as separable equivariant spatial and temporal modules to process spatio-temporal information.
They evaluate the proposed method on different benchmarks and vastly outperform frame-to-frame equivariant methods and non-equivariant spatio-temporal GNNs.

**Strengths:**

The paper studies the non-Markovian dynamics in physical systems, an often overlooked yet very important property.
It is well-written and easy to follow. The novelties are clear, and the ablation studies support their usefulness.
The quantitative results show a massive performance gain from incorporating equivariance and sequence dynamics.


**Weaknesses:**

The paper claims that "we are the first to use equivariant spatio-temporal graph models for physical dynamics simulation", yet it is missing out on a few related works, namely LoCS [1], and more recently, EqMotion [2]. Both works propose equivariant graph networks and focus on sequence-to-sequence prediction for physical systems. Hence, the authors should do a more thorough literature review and adjust their claims.

Many of the neural network modules used in the proposed method are not adequately described in the manuscript, and their significance is not tested with an ablation study. For example, the learnable parameters $w_k$ are only briefly described, and their exact form, as well as their usefulness, are unclear.

#### References
[1] Kofinas, Miltiadis et al. Roto-translated Local Coordinate Frames for Interacting Dynamical Systems. NeurIPS 2021.

[2] Xu, Chenxin et al. EqMotion: Equivariant Multi-agent Motion Prediction with Invariant Interaction Reasoning. CVPR 2023.

**Questions:**

1. Following the weaknesses above, a comparison with other spatio-temporal equivariant graph networks would further enhance the credibility of the proposed method.

2. How important is the use of $w_k$, given that these features are further processed during message passing?

3. Since this work focuses on non-Markovian dynamics, an ablation study on the optimal number of past timesteps would be beneficial and insightful.

**Limitations:**

The authors have adequately addressed the limitations.

---

> ### Author Rebuttal · Authors · 2023-08-10
>
> We are grateful for your positive and constructive comments, and provide the answers to your questions below.
> >Q1: Following the weaknesses above, a comparison with other spatio-temporal equivariant graph networks would further enhance the credibility of the proposed method.
>
> Thank you very much for raising these two papers: LoCS and EqMotion, which will be definitely cited and discussed in Related Work.  LoCS proposes two versions: the Markovian one and the non-Markovian one; the non-Markovian version is related to our method and it resorts to GRU units to record the memory of past frames, and predict the next frame conditional on the current frame in an auto-regressive manner, which can be regarded as the RNN style. On the contrary, our method resorts to the spatio-temporal setting and employs all past frames as the input to predict the target one, which can be regarded as the Transformer style. As for EqMotion, it first distills the input trajectory of each node into one multi-dimension vector, by which the spatio-temporal graph is compressed as one single spatial graph. By contrast, our method retains all input frames within both the spatial and temporal modules, such that it is able to better capture spatio-temporal correlations in a more elaborate way.
>
> Here, we additionally implement  EqMotion on MD17 and Motion datasets for evaluation, since its code is more friendly to be adjusted for these two tasks. For fair comparisons, the input of EqMotion only contain node coordinates, as the same as our method and other baselines.  We find that EqMotion performs much worse by directly predicting the absolute coordinates. We then modify EqMotion to predict the relative coordinates across two adjacent frames. Besides, we perform zero-mean normalization by subtracting all coordinate vectors of all nodes and all graphs with their mean, for further improvement of EqMotion on Motion dataset. The results are reported as follows, where the clear superiority of our ESTAG is still observed.
>
> |MD17 | ASPIRIN | BENZENE | ETHANOL | MALONALDEHYDE | NAPHTHALENE | SALICYLIC | TOLUENE | URACIL |
> |:---:|:---:|:---:|:---:|:---:|:---:|:---:|:---:|:---:|
> |EqMotion  | 0.721 | 0.156 | 0.476 | 0.600 | 0.747 | 0.697 | 0.691 | 0.681 |
> | ESTAG | 0.063 | 0.003 | 0.099 |0.101 | 0.068 | 0.047 | 0.079 | 0.066|
>
>
> | Motion | walk ($\times 10^{-1}$) | basketball ($\times 10^{-1}$) |
> |:---:|:---:|:---:|
> | EqMotion | 201.008 | 1362.900 |
> | EqMotion (zero-mean) | 1.011 | 4.893 |
> | ESTAG | 0.040 | 0.746|
>
> We will add the above discussions and adjust our claims accordingly.
>
> >Q2: How important is the use of $w_k$ given that these features are further processed during message passing?
>
> Sorry for the currently insufficient explanation. As mentioned Line 140-142, $w_k$ acts like spectral filters of the $k$-th frequency and enables us to select related frequency for the prediction. It is calculated as $\omega_k=f(\mathbf{h})$,
> where $w_k$ is a scalar ranging from 0 to 1, $f$ is implemented as an MLP plus a Sigmoid output, and $\mathbf{h}$ is the input feature. Here, we conduct an ablation study to evaluate the effect of $w_k$:
> |MD17 | ASPIRIN | BENZENE | ETHANOL | MALONALDEHYDE | NAPHTHALENE | SALICYLIC | TOLUENE | URACIL |
> |:---:|:---:|:---:|:---:|:---:|:---:|:---:|:---:|:---:|
> | ESTAG |0.063 | 0.003 | 0.099 | 0.101 | 0.068|0.047 | 0.079 |0.066 |
> |w/o $w_k$ | 0.071 | 0.003 | 0.102 | 0.104 | 0.081 | 0.081 |0.079| 0.069 |
>
> In a general sense, the use of $w_k$ can further promote the performance, although the improvement is not so remarkable.  We will include the above explanations into the revised paper.
>
> >Q3: Since this work focuses on non-Markovian dynamics, an ablation study on the optimal number of past timesteps would be beneficial and insightful.
>
> Thoughtful viewpoint! The number of past timesteps is indeed a significant factor we need to focus on and we did conduct an ablation study in Table 1 and Figure 3 in the supplementary material. We find that extending the number of past timesteps from 3 to 10 is able to generally improve the performance, which exhibits the necessity of non-Markovian modeling. We will highlight this point in the main paper.

---

> > ### Comment · Reviewer_Gpx3 · 2023-08-17
> > **Official Comment by Reviewer Gpx3**
> >
> > I would like to thank the authors for their rebuttal; they have addressed my questions and concerns. I am still positive towards this paper and I think its contribution is significant. I am keeping my score to 6. The authors should include the related works discussed during the rebuttal in the camera ready version, along with any experiments that compare against them.

---

> > > ### Author Response · Authors · 2023-08-17
> > > **Thanks for your feedback**
> > >
> > > Dear reviewer,
> > >
> > > Thank you very much for approving our work. We will include the related works discussed during the rebuttal, along with the relevant experiments, in the camera-ready version.

---

### Official Review · Reviewer_Hnbu · 2023-07-07

**Soundness:** 3 good
**Presentation:** 3 good
**Contribution:** 3 good
**Rating:** 7
**Confidence:** 3

**Summary:**

This work proposes a novel architecture for predicting physical dynamics. This architecture first extracts the frequency feature of the input dynamics using a new technique. The frequency feature is then processed by spatial-temporal networks to generate predictions for the future dynamic. Through evaluating the novel architecture on multiple datasets ranging from molecular to macro levels, the authors show that their architecture outperform existing models significantly, especially on the molecular benchmark. The authors also present ablation studies to show which module is the most critical in yielding better performance.

**Strengths:**

The architecture proposed in this work (ESTAG) is novel. It is also shown to significantly outperform earlier works in this work on multiple datasets. The paper is well written and easy to read. The ablation studies conducted in this work also shows that the newly proposed frequency computation technique seems to be the most important module in the model. The result on the molecular dataset (MD17) is especially convincing that ESTAG is better than other models.

**Weaknesses:**

There are some details in the evaluation that are not clear. In the MD-17 dataset, the visualization seems to be really hard to differentiate the ESTAG model and the STEGNN model. But why is the MSE difference so big? Is the observed MSE difference in fact important for whatever downstream task that’s important? Or the STEGNN’s result is already good enough? The other two datasets (Protein and Motion datasets) show that ESTAG is still better than other models, but the gap is much smaller than the MD-17 dataset. It is unclear to me why ESTAG on MD-17 yields such a good result but not on others, can the authors provide an explanation on this? Moreover, on the motion dataset, only one person’s trajectory is used. Why only using one person’s trajectory but not all subjects’?  Why visualizations on the other two datasets are not provided?

**Questions:**

See the weakness.

**Limitations:**

yes

---

> ### Author Rebuttal · Authors · 2023-08-10
>
> We are grateful for your positive and constructive comments, and provide the answers to your questions below.
>
> >Q1：In the MD17 dataset, the visualization seems to be really hard to differentiate the ESTAG model and the STEGNN model. But why is the MSE difference so big? Is the observed MSE difference in fact important for whatever downstream task that’s important? Or the STEGNN’s result is already good enough?
>
> Sorry for the unclear visualization showing the difference between ESTAG and STEGNN in Figure 4. Actually, for the molecules (ASPIRIN, NAPHTHALENE, SALICYLIC) highlighted by red rectangles, the difference between  ESTAG and STEGNN is obvious.  For those nonobious cases, the unclarity is mainly caused by 2D printing that is hard to depict 3D conformations such as angle deviation. For better visualization, we choose the molecule URACIL which is not clearly visualized in the paper, and redisplay its 3D atom coordinates in Figure A4 (General Response). ESTAG yields more accurate 3D prediction than STEGNN, which is consistent with the MSE difference. We will add the new visualizations to the revised paper.
>
> >Q2: The other two datasets (Protein and Motion datasets) show that ESTAG is still better than other models, but the gap is much smaller than the MD17 dataset. It is unclear to me why ESTAG on MD17 yields such a good result but not on others, can the authors provide an explanation on this?
>
> Thank you for this nice observation. Here, we provide some potential explanations on why ESTAG on MD17 yields such a good result but not on the other two datasets:
> 1. On Protein dataset, the dynamics of a protein is much more complicated than those small molecules on MD17, owing to various kinds of physical interaction between different amino acids, let along each amino acid compose of a certain number of atoms.  We conjecture that our ESTAG is still hard to reveal sufficiently accurate dynamical patterns for proteins, even though its performance is already better than other methods.
> 2. On Motion dataset, ESTAG performs much better than other methods except STGCN. We suspect that the simulation of the walking motion is not that challenging. When we follow the reviewer's suggestion in Q3 and additionally conduct evaluation on a more challenging and complicated task: the basketball motion, the gap between ESTAG and STGCN is significant (see the table below), which indicates the effectiveness of ESTAG in broad and practical cases.
>
> | Motion_basketball | MSE ($\times 10^{-1}$) |
> |:---:|:---:|
> | PT-s | 886.023 |
> | PT-m | 413.306 |
> | PT-t | 15.878 |
> | Baseline-s | 749.486 |
> | Baseline-m | 335.002 |
> | Baseline-t | 12.492 |
> | GNN | 15.336 |
> | EGNN | 13.199 |
> | TFN | 13.709 |
> | SE3 | 13.851 |
> | STGCN | 4.919 |
> | ESTAG | 0.746 |
>
> We will add the above explanastions to the revised paper.
>
> >Q3: Moreover, on the motion dataset, only one person’s trajectory is used. Why only using one person’s trajectory but not all subjects’?
>
> Thank you for this comment. Indeed, there are more than one trajectories in one subject. The reason why we only use one subject (#subject 35) for evaluation is following the setting of GMN [16] which is the initial work to explore equivariant dynamics simulation on this motion dataset. To better demonstrate the effectiveness of our method in more cases, we additionally carry out experiments on the basketball motion (#subject 102) which is more challenging to simulate. Notably, for basketball motion data, we focus on the trajectories whose length is greater than 170. The results are reported when answering Q2, where we can observe that our ESTAG outperforms other methods remarkably. Conducting evaluations on all subjects (the total number is 144) is expensive and could be better left for future exploraion.
>
> >Q4: Why visualizations on the other two datasets are not provided?
>
> We apologize for the missed visualization in the paper. Here, we redisplay the visualization on Protein and Motion Cature in Figure A1-A3 (General Response). It can be seen that ESTAG achieves better prediction accuracy.

---

### Official Review · Reviewer_iyxR · 2023-07-09

**Soundness:** 3 good
**Presentation:** 3 good
**Contribution:** 2 fair
**Rating:** 5
**Confidence:** 4

**Summary:**

This paper aims to simulate the physical dynamics with a spatio-temporal attentive graph network. The major contribution is to integrate the concept of spatio-temporal graph neural network with DFT to capture the data dependencies. The proposed model is evaluated on three datasets regarding the molecular-, protein- and macro-level prediciton.

**Strengths:**

1. The paper is well-written and easy to follow.
2. The proposed model is technically solid.
3. The experiments cover three real-world datasets.

**Weaknesses:**

1. My primary concern is technical novelty. This paper adapts STGNN for physical dynamic simulation without significant contributions or novel designs. The paper claims that it is important to relieve from the Markovian assumption, however, it has been widely explored in the literature on STGNNs.
2. The related work should be further investigated. As far as I know, there are many advanced STGNNs that can achieve much higher accuracy than STGCN (which was proposed in 2018). For more details, please refer to a recent survey [1].
3. More powerful baselines should be discussed and considered as baselines [1]. This paper only compares ESTAN with STGCN in the line of STGNNs, which is not convincing enough. For example, comparing ESTAG with an existing STGNN published in 2022 is not investigated.

Reference:
[1] Jin, Guangyin, et al. "Spatio-temporal graph neural networks for predictive learning in urban computing: A survey." arXiv preprint arXiv:2303.14483 (2023).

**Questions:**

Please see the weaknesses.

---

> ### Author Rebuttal · Authors · 2023-08-10
>
> Thank you for your comments!  We provide the following responses to your concerns:
> >Q1: My primary concern is technical novelty. This paper adapts STGNN for physical dynamic simulation without significant contributions or novel designs. The paper claims that it is important to relieve from the Markovian assumption, however, it has been widely explored in the literature on STGNNs.
>
> The reviewer probably misunderstood our contributions. We strongly disagree that our paper adapts STGNN without significant contributions or novel designs. We provide the reasons below.
>
> 1. **Equivariance is not well explored in most STGNNs.**
> The main focus of our paper is on the task of 3D physical dynamics simulation, while most spatio-temporal GNNs discussed in the mentioned survey [A] are developed for uban computing. The unique challenge of physical dynamics simulation compared to uban computing is that, the model for physical dynamics simulation should obey E(3) equivariance: transforming the input coordinates by any translation/rotation/reflection will result in the output transformed in the same way. Most STGNNs, unfortunately, do not satify this crucial symmetry, as already pointed out in Line 50-52 in the paper. Exploring equivariant GNNs is currently a popular and challenging topic in machine learning [24, 16]. Our paper moves a step forward by investigating equivariant spatio-temporal GNNs, which exhibit particular difficulty and untriviality; for example, we should make both the spatial and temporal message passing equivariant.
> We understand that the non-Markovian property can be modeled in STGNNs, but our claim of relieving from the Markovian assumption is made in comparison with those equivariant GNNs without spatio-temporal modeling (such as EGNN). Overall, we propose an equivariant version of spatio-temporal GNNs, which can not only encode the  non-Markovian property but also ensure E(3) equivariance for physical simulation.
>
> 2. **The proposed equivariant model is novel.**
> The entire architecture we design is novel, which consists of three equivariant modules: Equivariant Discrete Fourier Transform (EDFT),  Equivariant Spatial Module (ESM), and Equivariant Temporal Module (ETM). Particularly, to the best of our knowledge, there is no previous attempt to develop equivariant DFT; in this paper, we achieve this by first translating the signals by the mean position and then adopt the same basis over the spatial dimension (Eq.3). The extracted invariant frequencies are then embedded into ESM to better leverage periodicity patterns. For ETM, the equivariant attention-based mechanism is also novel and carefully designed to ensure equivariance.
>
> >Q2: The related work should be further investigated. As far as I know, there are many advanced STGNNs that can achieve much higher accuracy than STGCN (which was proposed in 2018). For more details, please refer to a recent survey [A].
>
> Thank you for your suggestion. We will cite the mentioned survey in Related Work. Indeed, besides STGCN [34], we did investigate ST-GCN [33], GaAN [35], and ASTGCN [13] in Line 85-95. We will discuss more advanced methods as suggested. It is worth mentioning that most spatio-temporal GNNs mentioned in [A] focus on urban computing, while our paper is concerned with the task of 3D physical dynamics simulation.
>
> >Q3: More powerful baselines should be discussed and considered as baselines [A]. This paper only compares ESTAG with STGCN in the line of STGNNs, which is not convincing enough. For example, comparing ESTAG with an existing STGNN published in 2022 is not investigated.
>
> Thank you for your suggestion. We have carefully read the survey you mentioned and choose AGL-STAN published in 2022 for comparison, since the encoder it used is transformer-based and competitive in performance. We find that AGL-STAN is originally for crime prediction, and it performs much worse for our task by directly predicting the absolute coordinates (i.e. AGL-STAN (abs)). We then modify AGL-STAN to predict the relative coordinates across two adjacent frames (i.e. AGL-STAN (rel)). The results become better and are tabulated below:
> |MD17|ASPIRIN|BENZENE|ETHANOL|MALONALDEHYDE|NAPHTHALENE|SALICYLIC|TOLUENE|URACIL|
> |-|-|-|-|-|-|-|-|-|
> |AGL_STAN (abs)|4.084|1.651|1.358|1.135|0.938|1.003|1.502|0.784|
> |AGL_STAN (rel)|0.719|0.106|0.459|0.596|0.601|0.452|0.683|0.515|
> | ESTAG | 0.063 | 0.003 |0.099 | 0.101 | 0.068 | 0.047 | 0.079 | 0.066 |
>
>
> |Protein|MSE|
> |-|-|
> |AGL_STAN(abs)|1.859|
> |AGL_STAN(rel)|1.671|
> | ESTAG |1.471 |
>
>
> |Motion|walk($\times 10^{-1}$)|basketball($\times 10^{-1}$)|
> |-|-|-|
> |AGL_STAN (abs)|1.675|189.082|
> |AGL_STAN (rel)|0.037|5.734|
> | ESTAG | 0.040 |0.746|
>
> We still observe that our ESTAG generally outperforms AGL-STAN. AGL-STAN (rel) performs slightly better than ESTAG on Motion-Walk, but for Motion-Basketball which is more complicated to simulate, ESTAG yields a much lower MSE than AGL-STAN (0.746 vs 5.734).  Again, we highlight that AGL-STAN is not equivariant, and it could fail if we steer the input via E(3) transformation. The above discussions will be added into the revised paper to address the reviewer's concern.
>
> [A]  Jin, Guangyin, et al. "Spatio-temporal graph neural networks for predictive learning in urban computing: A survey."

---

> ### Comment · Reviewer_iyxR · 2023-08-17
> **Response**
>
> Dear authors,
>
> Thank you for addressing my concerns. I also read the comments from other reviewers. Personally, I still have reservations regarding the technical novelty of the paper in the context of STGNNs. Since other concerns have been resolved, I would like to raise my recommendation score after consideration.
>
> Best regards,
> Reviewer

---

> > ### Author Response · Authors · 2023-08-17
> > **Thanks for your response**
> >
> > Dear reviewer,
> >
> > we sincerely appreciate your recognition of our efforts. We will further clarify our contributions and the techinical novelty in the revised version.

---

### Author Rebuttal · Authors · 2023-08-10

## General Response

We sincerely thank all reviewers and ACs for their time and efforts on reviewing the paper. We are glad that the reviewers recognized the contributions of our paper, which we briefly summarize as follows.
- **Novelty**. "The architecture proposed in this work (ESTAG) is novel" (Hnbu); "The novelties are clear" (Gpx3); "This approach is well-founded" (pTt6); "The idea presented in the paper is novel" "It fills a gap in the existing framework and the direction is practical and meaningful" (TWyG).
- **Presentation**. "The paper is well-written and easy to follow" (iyxR); "The paper is well written and easy to read" (Hnbu); "It is well-written and easy to follow" (Gpx3);“The illustrations are very easy to follow” (TWyG).
- **Experiment**. "The experiments cover three real-world datasets" (iyxR);  "It is also shown to significantly outperform earlier works in this work on multiple datasets" (Hnbu); "The quantitative results show a massive performance gain from incorporating equivariance and sequence dynamics" (Gpx3);  "Comprehensive experimental validation" (pTt6); “There are empirical improvements” (TWyG).

We also appreciate the reviewers for their thoughtful comments and concerns, and provide additional visualizations and experment results in the attached PDF file for more details. We summarize the extra contents as follows.

- **Figure A1 (to Reviewer Hnbu)** visualizes the difference between ESTAG and ST-EGNN on the walk subject on Motion dataset, showing that ESTAG ahieves clearly better prediction accuracy.
- **Figure A2 (to Reviewer Hnbu)** visualizes the difference between ESTAG and ST-EGNN on the basketball subject on Motion dataset, again showing that ESTAG ahieves clearly better prediction accuracy.
- **Figure A3 (to Reviewer Hnbu)** visualizes the difference between ESTAG and ST-EGNN on Protein dataset, where the protein predicted by ESTAG aligns closer to the ground truth while ST-EGNN fails to predict the alpha helix.
- **Figure A4 (to Reviewer Hnbu)** visualizes  the 3D conformation of the molecular URACIL on MD17, where our ESTAG yields closer prediction to the ground truth.
- **Figure A5 (to Reviewer pTt6)** visualizes rollout MSE on MD17, verifying the effectiveness of ESTAG in the recurrent forecasting scenarios.

---

### Decision · Program_Chairs · 2023-09-21

**Decision:**

Accept (poster)

**Comment:**

The paper discusses the problem of modelling non-markovian dynamics in dynamical systems, and specifically by making sure the model preserves equivariance. All reviewers agree that this is a sufficiently novel paper with a good set of experiments. All criticisms were well addressed and reviewers even raised their scores. All in all, I think it is a clear accept.